# Regulatory dissection of the severe COVID-19 risk locus introgressed by Neanderthals

Evelyn Jagoda[1†], Davide Marnetto[2†], Gayani Senevirathne[1], Victoria Gonzalez[3,4], Kaushal Baid[4], Francesco Montinaro[2,5], Daniel Richard[1], Darryl Falzarano[3,4], Emmanuelle V LeBlanc[6], Che C Colpitts[6], Arinjay Banerjee[3,4,7,8], Luca Pagani[2,9], Terence D Capellini[1,10]*

[1]Department of Human Evolutionary Biology, Harvard University, Cambridge, United States; [2]Estonian Biocentre, Institute of Genomics, University of Tartu, Tartu, Estonia; [3]Department of Veterinary Microbiology, University of Saskatchewan, Saskatoon, Canada; [4]Vaccine and Infectious Disease Organization, University of Saskatchewan, Saskatoon, Canada; [5]Department of Biology, University of Bari, Bari, Italy; [6]Department of Biomedical and Molecular Sciences, Queen's University, Kingston, Canada; [7]Department of Biology, University of Waterloo, Waterloo, Canada; [8]Department of Laboratory Medicine and Pathobiology, University of Toronto, Toronto, Canada; [9]Department of Biology, University of Padova, Padova, Italy; [10]Broad Institute of MIT and Harvard, Cambridge, United States

*For correspondence:
tcapellini@fas.harvard.edu

[†]These authors contributed equally to this work

Competing interest: The authors declare that no competing interests exist.

**Abstract** Individuals infected with the *SARS-CoV-2* virus present with a wide variety of symptoms ranging from asymptomatic to severe and even lethal outcomes. Past research has revealed a genetic haplotype on chromosome 3 that entered the human population via introgression from Neanderthals as the strongest genetic risk factor for the severe response to COVID-19. However, the specific variants along this introgressed haplotype that contribute to this risk and the biological mechanisms that are involved remain unclear. Here, we assess the variants present on the risk haplotype for their likelihood of driving the genetic predisposition to severe COVID-19 outcomes. We do this by first exploring their impact on the regulation of genes involved in COVID-19 infection using a variety of population genetics and functional genomics tools. We then perform a locus-specific massively parallel reporter assay to individually assess the regulatory potential of each allele on the haplotype in a multipotent immune-related cell line. We ultimately reduce the set of over 600 linked genetic variants to identify four introgressed alleles that are strong functional candidates for driving the association between this locus and severe COVID-19. Using reporter assays in the presence/absence of *SARS-CoV-2*, we find evidence that these variants respond to viral infection. These variants likely drive the locus' impact on severity by modulating the regulation of two critical chemokine receptor genes: *CCR1* and *CCR5*. These alleles are ideal targets for future functional investigations into the interaction between host genomics and COVID-19 outcomes.

## Editor's evaluation

A genetic haplotype on chromosome 3 that entered the human lineage from mating with Neanderthals has previously been implicated as a strong genetic risk factor for severe COVID-19 outcomes. This study uses population genetics and functional genomics tools along with experimental assays to assess the genetic variants in these regions for their likelihood of driving the severe COVID-19 phenotype. They ultimately identify 4 (out of about 600) variants as strong functional

candidates. This study is a valuable contribution to the interaction between host genomics and COVID-19 outcomes and provides compelling evidence allowing for more targeted future functional investigations.

## Introduction

Since its emergence in late 2019, *SARS-CoV-2* has infected more than 160 million people worldwide and claimed more than 3 million lives (*WHO, 2021*). The variance in patient outcomes is extreme, ranging from no ascertainable symptoms in some cases to fatal respiratory failure in others (*Vetter et al., 2020*). This wide range of patient outcomes is due in part to comorbidities; however, prior health conditions do not explain the full range of outcomes (*Zhou et al., 2020*). Therefore, efforts have been made to assess a potential genetic component. Repeatedly, a region on chromosome 3 encompassing a cluster of chemokine receptor genes has been reported as having a strong association with an increase in COVID-19 severity in Europeans, with the strongest reported risk variant conferring an odds ratio of 1.88 for requiring hospitalization (p=2.7*10$^{-49}$, *COVID-19 Host Genetics Initiative, 2021*).

*Zeberg and Pääbo, 2020* identified that the strongest COVID-19 severity locus was introgressed by Neanderthals, with a core introgressed haplotype spanning ~49 kb from chr3:45,859,651–45,909,024 (hg 19) including rs35044562, reported as one of the leading variants of the association (*COVID-19 Host Genetics Initiative, 2020*) and a broader, extended haplotype with reduced linkage spanning ~333 kb from chr3:45,843,315–46,177,096. Subsequently, this locus was fine-mapped to two independent risk signals, one which is confirmed to fall within the Neanderthal haplotype and tagged by a set of strongly linked SNPs including rs35044562 and rs10490770, while the other, led by rs2271616, falls just upstream (*COVID-19 Host Genetics Initiative, 2021*; *Kousathanas et al., 2022*). The core haplotype is at highest frequency in South Asian populations (30%), as well as at appreciable frequency in Europe (8%) and the Americas (4%), yet it is virtually absent in East Asia. The stark difference in frequency between South Asian and East Asian populations implies that the haplotype may have been positively selected in South Asian populations, for which there is support (*Racimo et al., 2014*; *Jagoda et al., 2018*; *Browning et al., 2018*) and/or subject to purifying selection in East Asian populations. However, the specific phenotypic consequences of this haplotype leading to its potential adaptive effect as well as its effect on COVID-19 severity remain unknown. Moreover, the potential causal drivers of the selective pressure, as well as COVID-19 severity remain unstudied.

Here, we identify putative functional variants within this haplotype that may be driving its association with COVID-19 severity. To do so, we first examine the haplotype in the context of a broader introgressed segment. We then identify loci within the introgressed segment that are associated with levels of gene expression (eQTLs) in vivo. We next compare the eQTL effects of these variants with differentially expressed genes in COVID-19 and related infection datasets to identify which response genes for these eQTLs are potentially relevant to the COVID-19 phenotype. We follow this computational approach with a high-throughput functional Massively Parallel Reporter Assay (MPRA) and identify 20 variants along the introgressed segment that directly modulate reporter gene expression. We intersect these 20 variants with a host of molecular and phenotypic datasets to further refine them to 4 which display the strongest evidence of contributing to the genetic association with severe COVID-19 at this locus. We then investigated these four variants (eight alleles) using reporter assays in the context of the promoter of their most likely endogenous target gene (*CCR1* or *CCR5*), and in the presence/absence of replicating *SARS-CoV-2*, revealing evidence of important functionality. These tested variants primarily modulate expression through their potential effects on *CCR1* and *CCR5 cis*-regulation and are strong candidate variants that should be investigated with future targeted functional experiments. An overview of this experimental workflow is shown in *Figure 1*.

## Results

### Genome-wide scans for Neanderthal introgression

We carried out two genome-wide searches for introgressed loci in a European population for which we also had available eQTL data using Sprime (*Browning et al., 2018*) and U and Q95 (*Racimo et al., 2014*) methods. We used 423 Estonian whole genome samples (*Pankratov et al., 2020*) that

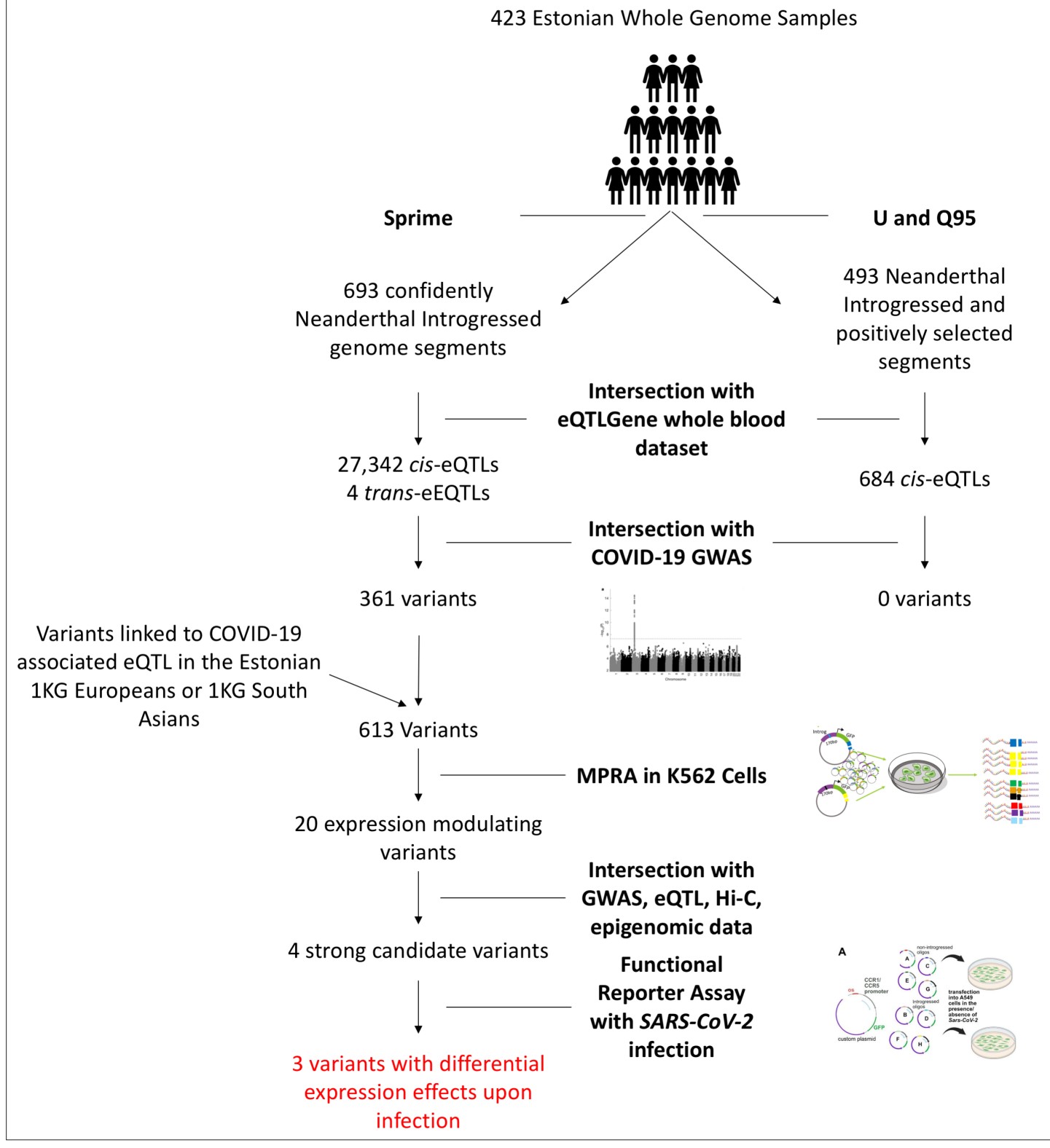

**Figure 1.** Overview of experimental workflow from whole genome scans for Neanderthal introgression to variant section for the MPRA and *SARS-CoV-2* infection reporter assay experiments.

constitute a well-studied representative sample of the broader Estonian population as sampled by the Estonian Biobank (EGCUT) (*Leitsalu et al., 2015*). These samples also have available whole blood RNA-sequence data which contributed to eQTLGen, a broad whole blood eQTL analysis study (*Võsa et al., 2018*). By utilizing genomes that were part of the eQTL study population, we can be assured that the associations between alleles and gene expression is accurate, as differential linkage disequilibrium (LD) between alleles in different populations can decrease the efficacy of using eQTL data from one population on another.

We initially conducted the Sprime scan (*Browning et al., 2018*) using the 423 Estonians as the ingroup population along with 36 African samples from the Simons Genome Diversity Project (SGDP) with no evidence of European admixture (*Mallick et al., 2016*) as an outgroup (*Supplementary file 1a*). From this scan, we identified 175,550 likely archaically introgressed alleles across 1,678 segments (*Supplementary file 1b*). Following *Browning et al., 2018*, we then identified segments as confidently introgressed from Neanderthals if they had at least 30 putatively archaically introgressed alleles with a match rate to the Vindija Neanderthal genome (*Prüfer et al., 2017*) greater than 0.6 and a match rate with the Denisovan genome (*Meyer et al., 2012*) less than 0.4 (*Browning et al., 2018*). In total, we identified 693 such segments (*Supplementary file 1c*), including the segment containing the COVID-19 severity haplotype on chromosome 3 (see above).

We next used the U and Q95 scan, which specifically identifies regions of introgression showing evidence of positive selection (*Racimo et al., 2017*). Using Africans from the 1000 genomes project as an outgroup (*Auton et al., 2015*), we found 493 such regions (*Supplementary file 1d*). We did not detect the introgressed COVID-19 severity haplotype in our population via this method. This suggests that the COVID-19 severity associated segment, while likely introgressed from Neanderthals based on its detection in our Sprime scan and via the work of others (*Zeberg and Pääbo, 2020*), was not under positive selection in the Estonian population. This is consistent with the previously reported lower frequency (8%) of the haplotype in Europeans relative to South Asian populations in which the haplotype is at higher frequency (30%) (*Zeberg and Pääbo, 2020*). However, U and Q95 scans do detect this region in South Asian populations (*Racimo et al., 2017*; *Jagoda et al., 2018*), supporting positive selection on this haplotype in South Asian, but not European populations.

We next examined which alleles in these putatively Neanderthal introgressed regions detected using these two genome-wide scans also are *cis-* and *trans*-eQTLs in the eQTLGen whole blood dataset (*Võsa et al., 2018*). From the U and Q95 data, we identified 684 *cis*-eQTLs across 250 40 kb windows (*Supplementary file 1e*). There were no *trans*-eQTLs detected in this set. From the Sprime data, we found 27,342 *cis*-eQTLs from 318 segments along with four *trans*-eQTLs from three segments (*Supplementary file 1f* and *Supplementary file 1g*).

## Refinement of the severe COVID-19 associated introgressed segment

In our Sprime scan, we identified an introgressed region containing the haplotype defined by *Zeberg and Pääbo, 2020* as both introgressed and associated with increased risk of COVID-19-severity. The overall introgressed region as detected in our Estonian population spans ~811 kb from chr3:45,843,242–46,654,616, encompasses 16 genes (*Figure 2A*), and ranks in the top 2% (ranked 21/1677) of Sprime detected segments based on likelihood of introgression and in the top 5% (58/1677) of Sprime segments based on length. Its extreme length provides additional support for the fact that it is introgressed and not likely a product of incomplete lineage sorting, which is detected as seemingly introgressed tracts of significantly shorter length (*Huerta-Sánchez et al., 2014*).

To examine how the introgressed segment may be affecting COVID-19 severity, we began by examining the LD structure within the segment and identified four major blocks defined as minimum pairwise LD between Sprime-identified variants within a block (min r2=0.34) (*Figure 2—figure supplement 1*). (Please note, *Figure 2* includes SNPs linked to Sprime variants whereas here we are exclusively discussing SNPs directly identified in the Sprime scan). We labeled these blocks as 'A' from rs13071258 to rs13068572 (chr3:45,843,242–46177096), 'B' from rs17282391 to rs149588566 (chr3:46,179,481–46,289,403), 'C' rs71327065 to rs79556692 (chr3:46,483,630–46,585,769), and 'D' from rs73069984 to rs73075571 (chr3:46593568–46649711) (*Figure 2A*; *Figure 2—figure supplement 1*). All Sprime alleles in the A block are significantly ($p<5*10^{-8}$) associated with increased risk for COVID-19 severity (*COVID-19 Host Genetics Initiative, 2021*), with the median p-value being $2.32 * 10^{-26}$ and median effect size being 0.42 (*Figure 2B*). The B block also harbors many alleles (81.2%)

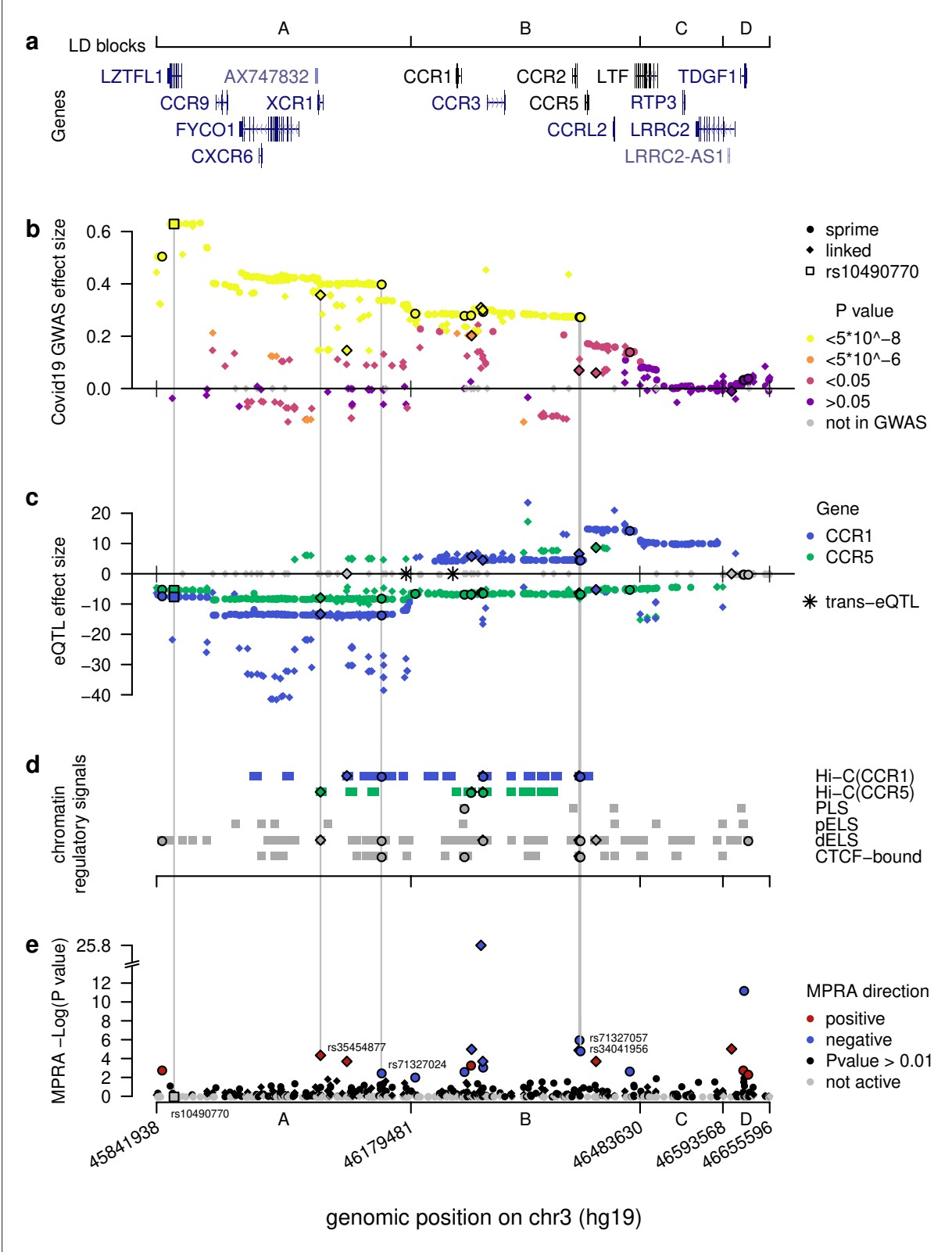

**Figure 2.** Computational intersections between MPRA emVars and functional genomics datasets across the severe COVID-19 risk locus. (**a**) Gene locations across the locus along with boundaries of the four LD blocks (**A–D**), borders extended to encompass all SNPs in LD (r2 > 0.3) tested with MPRA. (**b**) Severe COVID-19 GWAS effect sizes from release 5 of the COVID-19 Host Initiative dataset (2021), with strongest genome-wide p-values in yellow spanning the A and B blocks. See key for other color definitions. Dots and diamonds across the panels indicate respectively SNPs identified

*Figure 2 continued on next page*

*Figure 2 continued*

directly by Sprime (dots) and SNPs in linkage disequilibrium (r2 >0.3) with them (diamonds). (**c**) eQTL effect sizes across the locus (blue for *CCR1*, green for *CCR5*) in whole blood from eQTLGen (*Võsa et al., 2018*) across the locus. Note the strong down- versus up-regulation of *CCR1* for variants in the A versus B blocks, respectively. Grey SNPs are not eQTLs for any of the two genes or were absent from the eQTL study. Asterisks denote *trans*-eQTLs. (**d**) Chromatin-based functional annotations across the locus consisting of Hi-C contacts with *CCR1* and *CCR5* in Spleen, Thymus, or LCL (*Jung et al., 2019*) and candidate *cis*-regulatory elements from *Moore et al., 2020*. (**e**) -Log p-values for emVars identified using MPRA across the locus. Grey SNPs failed the test for activity in either the archaic or non-archaic form. Vertical lines connect the four putative causal emVars and the most cited tag SNP rs10490770 to functional genomics and genetics data. The four putatively causal variants are unique in having significant hits across all functional genomics and genetics tests.

The online version of this article includes the following figure supplement(s) for figure 2:

**Figure supplement 1.** Linkage Disequilibrium (LD) between Sprime identified introgressed variants within the segment (chr3:45,843,242–46,654,616) containing the COVID-19 associated haplotype.

**Figure supplement 2.** On the top panel we show UCSC Browser tracks for a 61 k region (chr3:45,849,651–45,911,089, hg19) encompassing the leading SNPs on the 3p21.31 COVID-19-risk-associated locus.

significantly associated with COVID-19 severity, with the median p-value being $1.94*10^{-9}$ and median effect size being 0.28 (*Figure 2B*). In the C and D block, no alleles are significantly associated with COVID-19 severity, suggesting that the most likely causal variants for the COVID-19 severity association are found within the A or B blocks (*Figure 2B*).

All the 361 Sprime-identified introgressed variants act as eQTLs in the whole blood (*Võsa et al., 2018*) including for many genes that are relevant to COVID-19 infection. Strikingly, of the four *trans*-eQTLs identified genome-wide in our Sprime scan regions in Estonians, two were located on the introgressed COVID-19 severity haplotype. These two variants, rs13063635 and rs13098911, have 11 and 33 response genes, respectively (*Supplementary file 1g*). We examined whether these response genes have any relevance to COVID-19 infection and found that 3 (27%) and 13 (39%) of the response genes for rs13063635 and rs13098911, respectively, are differentially expressed in at least one experiment in which a lung related cell-line or tissue was infected with COVID-19 or other related infections (*Supplementary file 1h* and *Supplementary file 1i*). These results suggest that these two *trans*-eQTLs may affect the lung response to COVID-19 in a way that could contribute to differential severity in host response. Furthermore, all 361 variants, including the two *trans*-eQTLs, act as *cis*-eQTLs in whole blood, altering the expression of 14 response genes: *CCR1, CCR2, CCR3, CCR5, CCR9, CCRL2, CXCR6, FLT1P1, LRRC2, LZTFL1, RP11-24F11.2, SACM1L, SCAP,* and *TMIE*. Of these genes, 7 are chemokine receptor genes (*CCR1, CCR2, CCR3, CCR5, CCR9, CCRL2,* and *CXCR6*), which are likely linked to the segment's association with COVID-19 severity.

Recent work has focused on pinpointing which of the aforementioned genes mediate(s) this risk signal, identifying *CXCR6* (*Schmiedel et al., 2021*; *Pairo-Castineira et al., 2021*; *Kasela et al., 2021*, *COVID-19 Host Genetics Initiative, 2021*), *CCR9* (*Schmiedel et al., 2021*; *Kousathanas et al., 2022*), *SLC6A20* (*Kasela et al., 2021*, *COVID-19 Host Genetics Initiative, 2021*; *Kousathanas et al., 2022*), *FYCO1* (*Schmiedel et al., 2021*), *LZFL1* (*Kousathanas et al., 2022*), and *CCR2* and *CCR3* (*Pairo-Castineira et al., 2021*) through bayesian fine-mapping, colocalization analyses, and transcriptome-wide association (TWAS). Although these studies focused on genes physically closer to the lead risk variant (rs10490770), epigenomic dissection and functional mapping also implicated *CCR1,2,3,5* genes (*Stikker et al., 2022*) which are farther but still deeply embedded in the introgressed haplotype. The association between COVID-19 phenotypes and *CCR1* and *CCR5* in particular also finds support from expression studies where elevated *CCR1* expression in neutrophils and macrophages has been detected in patients with critical COVID-19 illness (*Chua et al., 2020*), in biopsied lung tissues from COVID-19 infected patients (*Supplementary file 1h* and *Supplementary file 1i*), as well as in Calu3 cells directly infected with COVID-19 (*Supplementary file 1h* and *Supplementary file 1i*). Likewise, elevated *CCR5* expression has been detected in macrophages of patients with critical COVID-19 illness (*Chua et al., 2020*). Notably, some ligands for *CCR1* and *CCR5* (*CCL15, CCL2,* and *CCL3*) also show over-expression in these patients (*Chua et al., 2020*). *CCRL2, LZTFL1, SCAP,* and *SACM1L* are also differentially expressed in at least one experiment that measures differential expression of genes in lung tissues and related cell lines infected with COVID-19 or other viruses that stimulate similar immune responses (*Supplementary file 1h* and *Supplementary file 1i*).

Intriguingly, when considering the effect of the *cis*-eQTLs for *CCR1* across the entire segment, we find that the majority of alleles along the introgressed haplotype within the A block are associated with its down-regulation (average Z score = –12.3) (*Figure 2C*). On the other hand, the majority of alleles within the B and C blocks are associated with *CCR1* up-regulation (average Z scores = 7.1 and 10.2, respectively) (*Figure 2C*). It is important to note that these eQTL effects are determined based on whole blood from non-infected, healthy patients (*Võsa et al., 2018*). When considered in the context that severe COVID-19 phenotype is characterized by increased expression of *CCR1* (*Chua et al., 2020*), these risk-associated alleles having different directions of effect suggest that a complex change to the *CCR1* regulatory landscape driven by alleles across the introgressed segment may be contributing to the disease phenotype. When we consider *CCR5* expression, it shows a more consistent pattern in which the majority of alleles within the A-C blocks are associated with its down-regulation. This result is interesting as *CCR5* expression in patients with severe COVID-19 illness is higher than those with more moderate cases (*Chua et al., 2020*). However, given the strong LD within each of these segments, discerning the direct connection between one or more alleles driving these regulatory changes and the molecular and phenotypic signatures of severe COVID-19 remains difficult.

## MPRA variant selection and study design

To independently assess the regulatory impact of the alleles on this COVID-19 risk haplotype, we employed a Massively Parallel Reporter Assay (MPRA) to investigate which alleles on the introgressed haplotype directly affect gene expression. Alleles which have the ability to modulate gene expression in this reporter assay are candidate putatively functional alleles that may drive the association with COVID-19 severity by altering the expression of genes that facilitate the biological response to COVID-19. To ensure that we tested any potential risk variants on the haplotype, we included in the MPRA all variants directly identified in the Sprime scan as being within the introgressed COVID-19 severity associated segment (361), along with any allele linked ($r^2 >0.3$) to one of these Sprime alleles in the Estonian population (140 alleles) or any 1000 Genomes (*Auton et al., 2015*) European (150 alleles) or South Asian population (197 alleles). Therefore, here we are testing not only alleles on the introgressed haplotype that have a confirmed Neanderthal-specific origin, but also alleles along the introgressed haplotype that were either already present in the human population when the haplotype was introgressed, or arose anew in humans (i.e., human-derived alleles) on the introgressed haplotype following its introgression. After filtering for SNPs falling within simple repeat regions (*Benson, 1999*), which are not compatible with MPRA (*Tewhey et al., 2016*), we identified a total of 613 experimental variants. Of these variants, 293 are significantly ($p<5*10^{-8}$) associated with COVID-19 severity and another 15 approach significance ($p<5*10^{-6}$), whereas 118 were not tested in the original GWAS (*The COVID-19 Host Genetics Initiative Release 5*) (*Figure 2B*).

We conducted this assay in K562 cells, a leukemia cell line that displays multipotent hematopoietic biology, which allows for comparison between the MPRA data and the eQTLs identified on whole blood samples (see above). Furthermore, K562 cells can be induced into immune cell fates highly relevant to the COVID-19 severity phenotypes including monocyte, macrophage, and neutrophils (*Tabilio et al., 1983*; *Sutherland et al., 1986*; *Butler and Hirano, 2014*). Moreover, as K562 cells robustly grow and are transfectable using MPRA reagents, they permit the rapid, repeated acquisition of large numbers of cells, as observed in prior MPRAs (*Ulirsch et al., 2016*; *Ernst et al., 2016*). Finally, the availability of other published datasets generated on K562 cells (e.g., chromatin ChIP-seq data), allows for comparison between MPRA results, which are episomal in nature, and the endogenous behavior of the genome in the same cell type. However, we do note that MPRA results will also be limited as they will not directly reflect the response of alleles in the endogenous genome and within the in vivo tissues in which the COVID-19 response occurs. We therefore also integrated the MPRA results with datasets derived on endogenous immune tissues/cells to help improve our ability to identify biologically relevant candidate driver variants.

## MPRA reveals 20 expression modulating variants (emVars)

We built the MPRA library following Tehwey and colleagues (2016) and performed four replicates of the experiment in K562 cells (Methods). We observed that normalized transcript counts between replicates were highly correlated (Pearson's *R*>0.999 p=p-value<2.2e-16) (*Figure 3A*). As with other

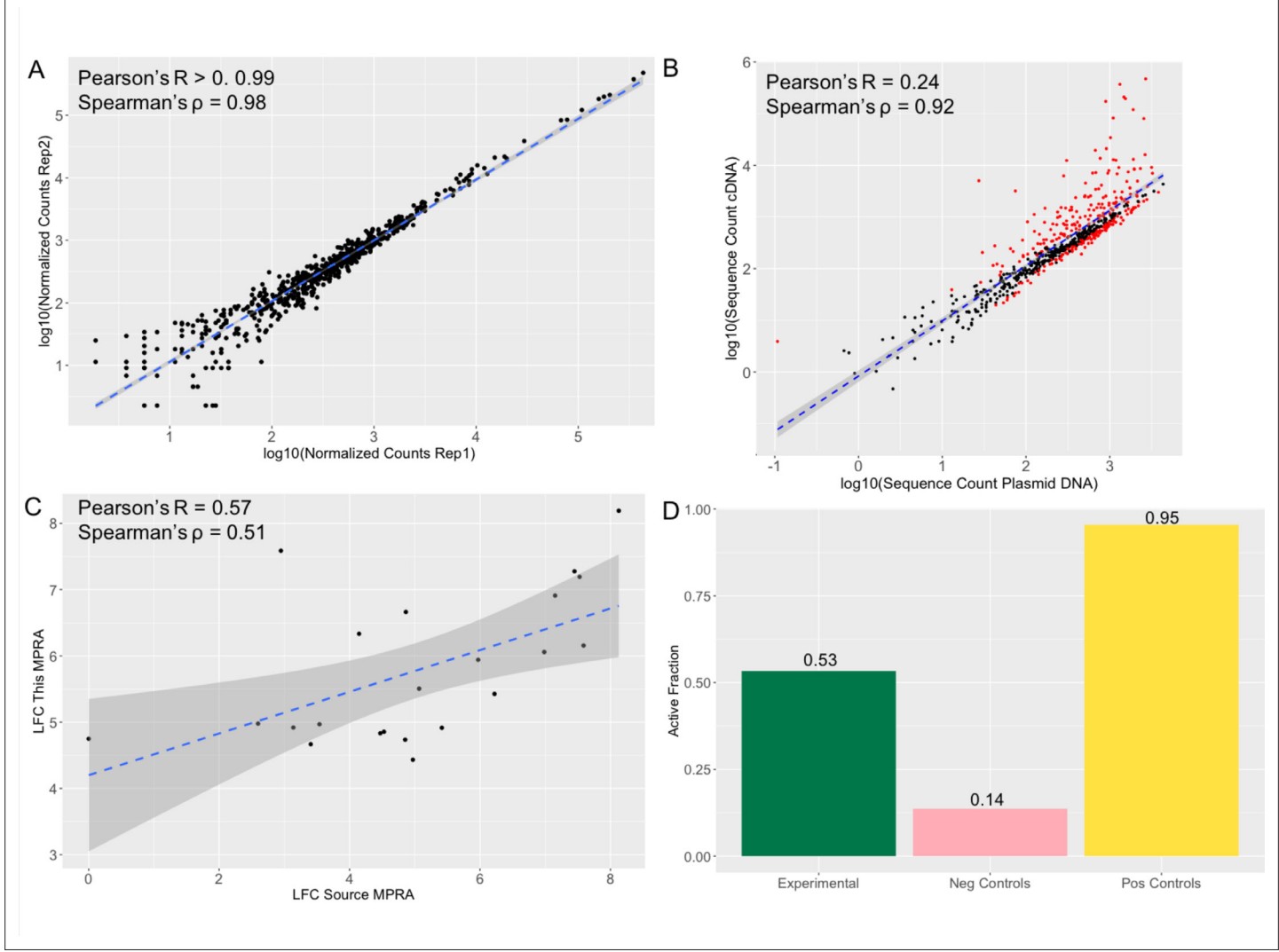

**Figure 3.** MPRA results show reproducibility and accuracy. (**a**) Log normalized counts for each tested sequence in replicate 1 compared with the replicate 2 of the MPRA. Pearson's R and Spearman's $\rho$ are extremely high and significant across pairwise replicate comparisons of all four replicates ($R$>0.99 p-value <2.2 *10$^{-16}$; $\rho$ =0.98 p-value <2.2 *10$^{-16}$). (**b**) Log normalized sequence counts for each tested in the plasmid DNA averaged across the four replicates compared with log normalized average sequence counts in the cDNA averaged across the four replicates. As with other MPRA studies (***Tewhey et al., 2016***; ***Uebbing et al., 2021***), there is a significant correlation but the plasmid counts do not fully explain the cDNA counts (Pearson's $R$=0.24 p-value <2.2 *10$^{-16}$; Spearman's $\rho$ =0.92 p-value <2.2 *10$^{-16}$), suggesting that some of the sequences have an effect on transcription. Sequences determined to be significantly active in the MPRA (methods) are colored in red, non-significant points are black. (**c**) Activity log fold change (LFC cDNA:pDNA) of positive control sequences in the source MPRA (***Jagoda et al., 2021***) and in this MPRA. The significant correlation (Pearson's $R$=0.57 p-value = 0.006; Spearman's $\rho$ =0.51 p-value 0.016) suggests that the activity results in this MPRA are accurate. (**d**) Fraction of sequences tested showing significant activity (LFC cDNA:pDNA corrected p-value >0.01). 95% of positive control sequences tested and 0.14% of negative control sequences tested show activity once again suggesting accuracy in the MPRA results. 53% of experimental sequences show significant activity.

MPRA studies (***Tewhey et al., 2016***; ***Uebbing et al., 2021***), transcript counts in the cDNA samples are significantly correlated with, but not completely explained by sequence representation in the DNA plasmid pool (correlation between means: Pearson's $R$=0.24 p=2.2e-16; Spearman's $\rho$ =0.912 p-value <2.2e-16) (***Figure 3B***), suggesting that while some sequences do not have an effect on transcription, other do. The expression of positive control sequences in this experiment was significantly correlated with their expression in the source MPRA (Pearson's $R$=0.59, p=0.0058) (***Figure 3C***). Any deviation between positive control sequence activity in this assay and the source MPRA for the two control sets is likely due to additional regulatory information in this assay for which the tested sequences are 270 bp compared with 170 bp in the source MPRA. Moreover, for our positive control set we observed

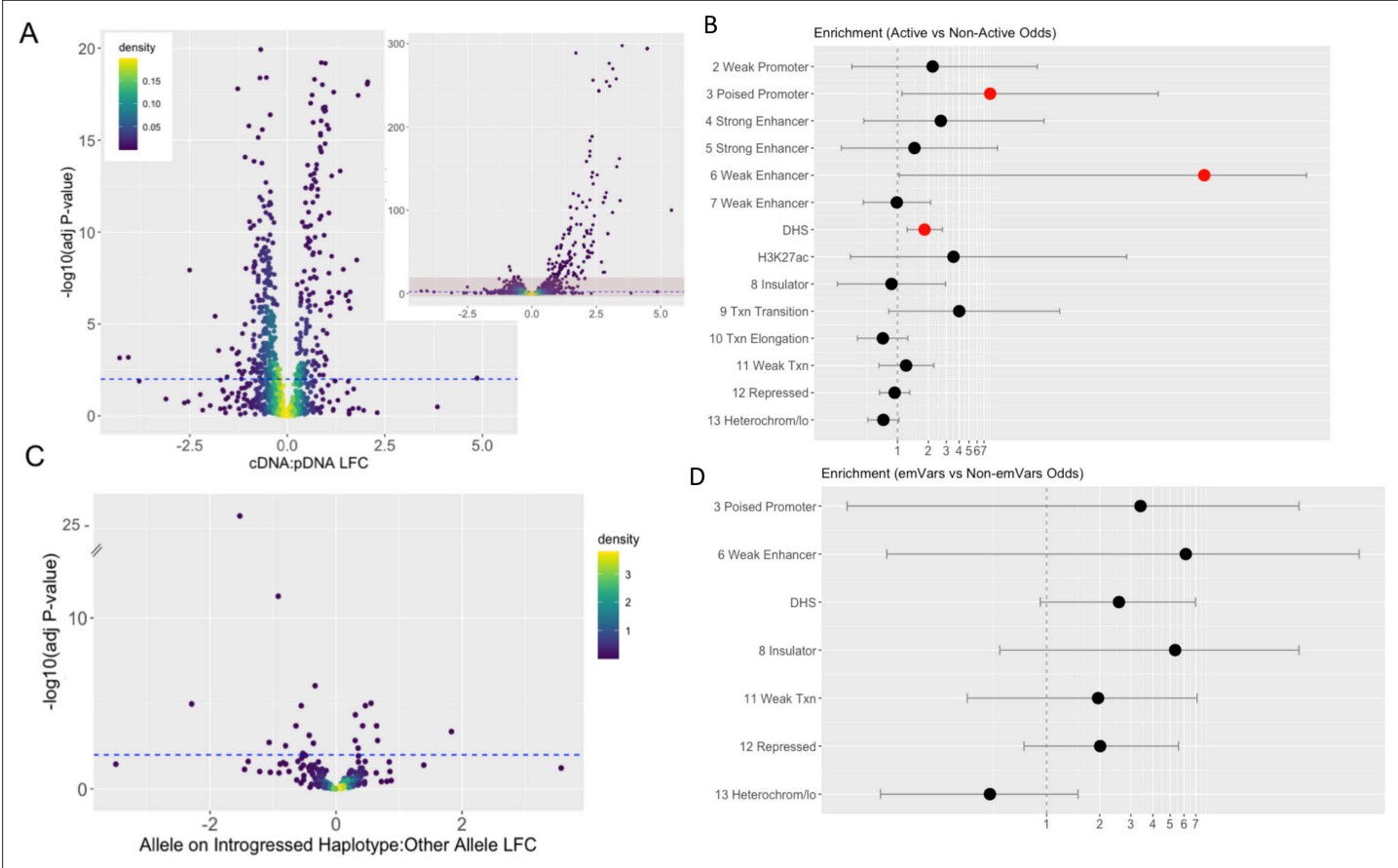

**Figure 4.** Properties of MPRA-identified active sequences and expression modulating variants. (**a**) Log Fold Change of the cDNA count compared to the plasmid DNA for each sequence and the -log$_{10}$ associated multiple hypothesis corrected p-value. Active sequences are those with a corrected p-value <0.01; this threshold is denoted with a blue dashed line. The larger plot has a y-axis limit of 20; the inset on the right shows the full spread of the data with the light red shaded box denoting the area shown in the larger figure to the left. (**b,d**) Enrichment of active sequences in K562 genomic features relative to non-active sequences (**b**) or sequences with emVars relative to sequences without emVars (**d**). Genomic features indicated with a number represent chromatin states in K562 cells as defined by *Ernst and Kellis, 2017*. DHS and H3K27ac derive from ENCODE (*Butler and Hirano, 2014*). Enrichments are reported as Fisher's odds ratios with lines indicating confidence intervals. Significant enrichments (p<0.05) are colored in red. Missing chromatin states had no overlap with either active sequences (**b**) or those containing emVars (**d**). (**c**) Log Fold Change between active sequences with the allele on the introgressed haplotype compared with the sequence containing the other allele. Expression modulating variants (emVars) are those whose LFC for this measure is significant with a corrected p-value <0.01; this threshold is denoted with a blue dashed line.

that 95% of control sequences displayed activity, whereas only 14% of the negative control sequences displayed activity (*Figure 3D*).

Of the 613 experimental (1,226 alleles) tested variants, 327 (53%) were within sequences found to have detected effects on reporter gene expression (i.e., they are considered 'active' or *cis*-regulatory elements [CREs]) in the context of either the allele on the introgressed haplotype or via its alternative variant (*Figure 3D*) (*Supplementary file 1j*). Consistent with other MPRA studies (*Tewhey et al., 2016*; *Ulirsch et al., 2016*; *Uebbing et al., 2021*), most active sequences showed relatively small effects, with only 17.1% of active sequences showing a log fold change (LFC) greater than 2 (*Figure 4A*). To confirm that these active CRE sequences reflect endogenous K562 biology, we compared the distribution of active CREs with K562 chromatin state data (*Ernst and Kellis, 2017*; *Sloan et al., 2016*). We observed that active CREs are significantly enriched relative to non-active sequences for falling with K562 DNase I Hypersensitivity Sites (DHS) and within poised promoters (OR: 8.05, p=0.023) (*Figure 4B*). They are also borderline significantly depleted of falling within heterochromatin (OR: 0.73, p=0.072) (*Figure 4B*).

We next defined as 'expression modulating variants' (emVars) those variants exhibiting a significant difference of expression between their two allelic versions using a multiple hypothesis corrected

p-value less than 0.01. Using this approach, we identified 20 emVars among the 613 variants we tested (*Table 1*, *Figure 2E*, *Figure 3B*). Consistent with previous MPRA studies (*Tewhey et al., 2016*; *Ulirsch et al., 2016*), the effect sizes of most emVars detected here are relatively modest, with only 1 emVar having an absolute LFC greater than 2 (*Table 1*, *Figure 4C*). Because the sample size is quite small (20), CREs containing emVars do not show significant enrichment within any endogenous K562 functional annotations. However, compared with tested sequences that do not contain emVars, CREs containing these emVars trend toward being over-represented within poised promoters, weak enhancers, DHS, insulators, repressive marks, and for depletion in heterochromatin regions (See *Figure 4D*).

### Evaluation of emVars with other functional data reveals four putatively causal variants

We next examined the 20 emVars for additional evidence of a role in the regulatory mechanisms potentially linked with COVID-19 response. Particularly, we looked for emVars that are (1) significantly ($p<5*10^{-8}$) associated with COVID-19 severity, (2) are eQTLs for a gene with strong evidence of relating to the severe COVID-19 phenotype, (3) are within a chromosomal region that physically interacts with the promoter of the COVID-19 associated gene in an immune cell line, according to Hi-C data, and (4) are within chromatin regions that have epigenomic marks consistent with acting as CREs in human immune cells (see 'emVar prioritization' section of Methods for full details).

Of the 20 emVars, we found four that meet all of these criteria (*Table 1*, *Figure 2*). Based on the combination of eQTL and Hi-C data, the variant rs35454877 is most likely implicated in the down-regulation of *CCR5*, while variants rs71327024, rs71327057, and rs34041956 appear to be involved in the regulation of *CCR1*. Of these, rs35454877 and rs71327024 fall within the LD Block A, which shows the strongest GWAS associations for COVID-19 severity and are 195 kb and 275 kb downstream of rs10490770, the most cited tag SNP for this GWAS signal (*COVID-19 Host Genetics Initiative, 2021*; *Schmiedel et al., 2021*). Furthermore, data from other GWAS studies shows that these four variants are significantly ($p<5*10^{-8}$) associated with other phenotypes that could relate to the COVID-19 severity phenotype including: 'Monocyte count', 'Granulocyte percentage of myeloid white cells', and 'Monocyte percentage of white cells' (*Astle et al., 2016*). Finally, all four SNPs above fall within ChIP-seq peaks for at least one transcription factor (TF) (*Supplementary file 1k*), with rs71327024 and rs35454877 overlapping peaks for as many as 89 and 36 different TFs, respectively (ENCODE TF ChIPseq NarrowPeaks from K562, GM12878 cell lines); among others, peaks for IKZF1, a key lymphocyte regulator (*Sellars et al., 2011*), are found overlapping all four SNPs. Nevertheless, we did not find that these SNPs alter TF binding affinity for any of the corresponding binding motifs analyzed (*Supplementary file 1l*). On the other hand, this analysis highlighted rs17713054, a SNP belonging to the top GWAS peak, where the archaic allele increases the binding affinity for CEBPB, a TF involved in Interleukin-mediated signaling (*Poli, 1998*). CEBPB was also independently shown to bind this region, a well-marked regulatory element, in two lung-derived cell lines (A549,IMR-90), see *Figure 2—figure supplement 2*. While rs17713054 was not classified as emVar in our experimental setting, further investigation on the role of this variant in modulating the response to *SARS-CoV-2* is warranted.

### Functional reporter analyses of top 4 emVars reveals causal variant activity in the presence of *SARS-CoV-2*

Chromatin Capture and eQTL data across various cell types led us to associate each of the four emVars to one of two target chemokine receptor genes, *CCR1* and *CCR5* (*Table 1*). We next sought to more directly connect these variants to expression of these two genes in the context of cells infected with *SARS-CoV-2*. Therefore, we designed a green fluorescent protein (GFP)-based reporter assay in which each allelic regulatory sequence (introgressed or non-introgressed emVar variant plus adjacent oligo-sequence) was cloned upstream of the promoter of the putative target gene (either *CCR1* or *CCR5*) and the coding sequence of GFP (*Figure 5A*). We transfected each reporter construct into human lung epithelial (A549) cells that were engineered to express the receptor of *SARS-CoV-2*, angiotensin-converting enzyme 2 (ACE2; *Blanco-Melo et al., 2020*; *LeBlanc and Colpitts, 2022*). Mock transfected or transfected A549-ACE2 cells were then mock infected or infected with *SARS-CoV-2* (see Methods; *Figure 5A*, *Figure 5—figure supplement 1*).

In the mock infected condition (i.e., absence of *SARS-CoV-2*), we found that construct pairs at emVar rs71327024 and at emVar rs35454877 exhibited statistically significant expression differences

**Table 1.** Properties of emVars and prioritization.

This table shows a summary of all the functional data we obtained on the 20 significant emVars identified by the MPRA which we used for prioritization of which these 20 emVars show the most evidence for contributing to the severe COVID-19 phenotype. Specifically, we looked for (1) concordance between the eQTL data (Columns H,I) and the Hi-C Data (K) - specifically for emVars that are eQTLs for a COVID-19 relevant gene with which they also physically interact in an immune tissue; (2) a significant association between the allele and severe COVID-19 in the GWAS data (column J); (3) overlap between the emVar and an ENCODE annotated cCRE (column L) with support in at least one 'class A' immune tissue (column M). The 4 variants that met all these criteria are highlighted in bold. For visualization of this data see *Figure 1*.

| snp | chr:pos_hg19 | Sapiens only Allele/ Allele on Introgressed Hap | Sprime (1)/ Linked (0) | LD Block | MPRA LFC (cDNA/ pDNA) | MPRA P-value | CCR1 eQTL Direction | CCR5 eQTL Direction | -log10(GWASp) | HI C Data (thymus, spleen, LCL) | cCREs | cCREs | cCREs Class A Immune Tissue |
|---|---|---|---|---|---|---|---|---|---|---|---|---|---|
| rs17712877 | 3:45848760 | G/C | 1 | A | 0.30 | 1.46E-03 | - | - | NA | x | dELS | | none |
| **rs35454877** | 3:46059484 | T/C | 0 | A | 0.31 | 4.60E-05 | - | - | 23.6 | FYCO1,CCR5 | dELS | | CD14-positive monocyte female donor ENCDO265AAA |
| rs35657218 | 3:46094462 | A/G | 0 | A | 0.64 | 2.01E-04 | 0 | 0 | 8.2 | CCR1,LZTFL1 | x | | x |
| **rs71327024** | 3:46140073 | G/T | 1 | A | -0.80 | 2.96E-03 | - | - | 24.7 | CCR1 | dELS,CTCF-bound | | CD14-positive monocyte female donor ENCDO265AAA, MM.1S, OCI-LY7, GM12878, PC-9, fibroblast of dermis, B cell adult |
| rs13083881 | 3:46184620 | C/G | 1 | B | -0.53 | 8.20E-03 | 0 | - | 10.3 | x | x | | x |
| rs34919616 | 3:46250008 | G/A | 1 | B | -0.36 | 2.11E-03 | 0 | - | 9.5 | x | PLS,CTCF-bound | | CD14-positive monocyte female donor ENCDO265AAA, MM.1S, GM12878 |
| rs35617677 | 3:46258740 | C/T | 1 | B | 1.83 | 4.38E-04 | 0 | - | NA | CCRL2,CCR5 | x | | x |
| rs34985947 | 3:46260008 | C/T | 0 | B | -2.29 | 1.06E-05 | + | - | 6.6 | CCRL2,CCR5,CCR2 | x | | x |
| rs1542755 | 3:46272440 | G/T | 0 | B | -1.53 | 1.68E-26 | 0 | - | 10.8 | CCR2 | x | | x |
| rs13096905 | 3:46274766 | G/A | 1 | B | -0.43 | 7.23E-04 | 0 | - | 9.3 | CCR5,CCR1,CCRL2,CCR2,CCR2 | dELS | | none |
| rs10510750 | 3:46274886 | T/C | 0 | B | -0.64 | 2.01E-04 | + | - | 9.5 | CCR5,CCR1,CCRL2,CCR2 | dELS | | none |
| rs762789 | 3:46402627 | G/A | 0 | B | 0.46 | 1.34E-05 | + | - | 1.6 | CXCR6,CCR1 | dELS,CTCF-bound | | CD14-positive monocyte female donor ENCDO265AAA |
| **rs71327057** | 3:46402645 | A/C | 1 | B | -0.33 | 9.19E-07 | - | - | 7.9 | CXCR6,CCR1 | dELS,CTCF-bound | | CD14-positive monocyte female donor ENCDO265AAA |

*Table 1 continued on next page*

*Table 1 continued*

| snp | chr:pos_hg19 | Sapiens only Allele/ Allele on Introgressed Hap | Sprime (1) / Linked (0) | LD Block | MPRA LFC (cDNA/ pDNA) | MPRA P-value | CCR1 eQTL Direction | CCR5 eQTL Direction | -log10(GWASp) | HI C Data (thymus, spleen, LCL) | cCREs | cCREs Class A Immune Tissue |
|---|---|---|---|---|---|---|---|---|---|---|---|---|
| rs34041956 | 3:46402734 | G/A | 1 | B | -0.55 | 1.34E-05 | + | - | 7.8 | CXCR6,CCR1 | dELS,CTCF-bound | CD14-positive monocyte female donor ENCDO265AAA |
| rs6777875 | 3:46425092 | G/T | 0 | B | 0.42 | 2.01E-04 | - | + | NA | x | dELS | B cell adult, MM.1S, OCI-LY7 |
| rs34245951 | 3:46469384 | T/C | 1 | B | -1.06 | 1.88E-03 | + | - | 2.7 | LZTFL1 | x | x |
| rs73072002 | 3:46605033 | C/T | 0 | D | 0.55 | 9.64E-06 | 0 | 0 | 0.1 | ALS2CL,LTF | x | x |
| rs55649212 | 3:46620020 | A/G | 1 | D | 0.66 | 1.46E-03 | 0 | 0 | 0.3 | x | x | x |
| rs2293021 | 3:46621117 | A/G | 1 | D | -0.92 | 5.35E-12 | 0 | 0 | 0.3 | x | x | x |
| rs73073752 | 3:46626599 | C/T | 1 | D | 0.35 | 4.07E-03 | 0 | 0 | 0.3 | x | dELS | none |

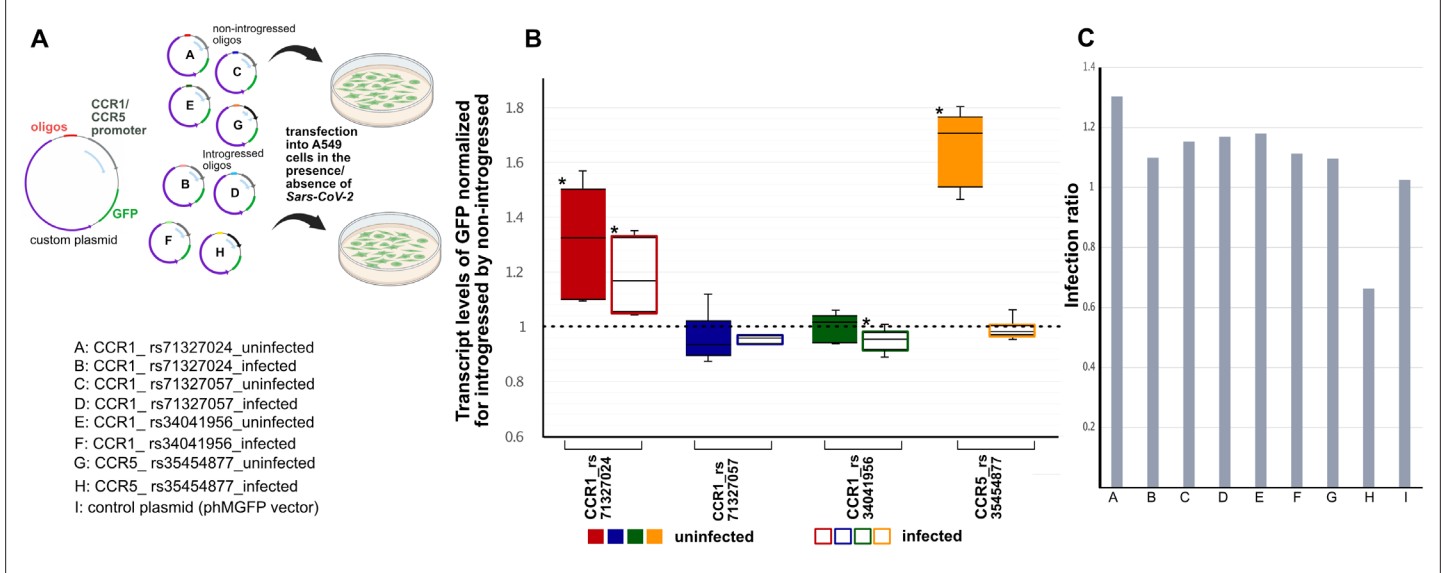

**Figure 5.** Activity of four emVars in *SARS-CoV-2* infected and uninfected A549-ACE2 cells using real-time PCR (qRTPCR). (**a, top**) Diagram of construct design and experimental transfection setup. (**a, bottom**) Constructs tested using in vitro transfection in A549-ACE2 cells whose results are shown in (**b**) in color-coded fashion. (**b**) Box-whisker plot graph depicting the transcript level of GFP expression driven by the introgressed allele normalized by the non-introgressed allele for each emVar in mock infection (i.e., absense of *SARS-CoV-2*) (left, solid color boxes) *vs SARS-CoV-2* infected (right, empty color boxes) cells. The '*' depicts significant changes using Wilcoxon statistical test that had a p-value of <0.05 (n=3 per condition). (**c**) Bar plots showing the infection ratio for each construct.

The online version of this article includes the following figure supplement(s) for figure 5:

**Figure supplement 1.** Transcript levels of experimental and control plasmids in the presence and absence of *SARS-CoV-2* infection.

between introgressed and non-introgressed alleles in A549-ACE2 cells, with higher levels driven by the introgressed alleles (i.e., above the dashed line at 1 in *Figure 5B*). Of the two, the response at emVar rs35454877 was the strongest, and was driven by increased activity by the introgressed allele. These findings in A549-ACE2 cells support our findings using MPRA, which was conducted using a minimal promoter in K562 cells.

However, in the presence of the *SARS-CoV-2*, we observed slightly different patterns (*Figure 5B*). Upon infection, we found that rs71327024 continued to be a significant emVar, with the introgressed allele driving 1.12 times more expression of GFP from the *CCR1* promoter than the non-introgressed allele. While we did not observe expression modulation for emVar rs304041956 in the mock infected state, we did now observe a modest effect size with the non-introgressed allele driving slightly more expression than the introgressed allele (0.9FC Introgressed: Non-introgressed). Interestingly, for emVar rs35454877, our strongest response eQTL in the mock infected state, we now observe no response between alleles, the marked drop resulting from the substantial decrease in the activity of the introgressed allele upon viral infection. This finding could indicate that the heightened effect of the introgressed allele in more normal (mock infected) settings is actively down-regulated in the context of viral infection settings (see Discussion).

## Discussion

We re-examined a previously identified adaptively introgressed segment in Eurasians within the context of the genome-wide signatures of introgression. We identified 613 variants within the introgressed region (chr3:45,843,242–46,654,616) and these were tested for whether they are potential drivers of the association this region exhibits with increased COVID-19 severity. Using MPRA, we tested these variants in a multipotent immune-related cancer cell line and narrowed down the list to 20 emVars where the expression level driven by the allele on the introgressed haplotype was significantly different from the expression driven by the other allele. We did not find support in the MPRA

for the expression modulation potential of rs10490770 (*COVID-19 Host Genetics Initiative, 2021*), rs35044562 (*Zeberg and Pääbo, 2020*), or rs11385942 (*Ellinghaus et al., 2020*), variants previously reported as tagging GWAS signals for COVID-19 severity. These variants may therefore be likely only tagging rather than causative, albeit functional experimentation in other cell lines and conditions may show otherwise. By further mining our MPRA results in concert with datasets on the epigenomic and transcriptional environment of immune cells from other functional genomics sources, here we highlight four emVars that have particularly strong evidence of acting as putatively causal variants and whose archaic alleles are strongly implicated with *CCR1* (rs71327024, rs71327057, rs34041956) and *CCR5* (rs35454877) regulation.

We next tested these four emVars in reporter assays in a lung cell line (A549-ACE2) capable of expressing ACE2, the receptor for *SARS-CoV-2*, and in the presence and absence of *SARS-CoV-2*. This allowed us to gain insight into how these emVars modulate target gene expression in a relevant COVID-19 cell type and infection state. Using this approach we whittled-down these emVars to three whose behavior changes in response to infection. Two emVars, emVar rs34041956 and emVar rs71327024, exhibited expression modulation between introgressed and non-introgressed variants in response to infection by the virus. While rs34041956 showed a very modest effect size, the introgressed allele at rs71327024 drove 1.12 times more expression of GFP driven by the *CCR1* promoter than the non-introgressed allele. Given that a consistent molecular symptom of severe COVID-19 response is elevated cytokine levels, particularly elevated *CCR1*, this variant is a prime causal variant for the association between this genetic haplotype and severe COVID-19.

Strikingly, one emVar, rs35454877 which had exhibited the strongest expression modulation in the absence of the virus, did not show allelic modulation upon infection. During infection, the introgressed allele at rs35454877 showed markedly reduced *CCR5* promoter driven reporter expression relative to the non-infection condition. Although this allele does not show modulation during infection, the 1.7x increase in GFP expression with a *CCR5* promoter that this allele drives in the non-infected state may translate to a baseline higher level of *CCR5* in individuals with this allele. Upon infection, the enhancer with the introgressed allele was severely down-regulated, to the level of the non-introgressed allele. Although the transcriptional down-regulation occurs during infection, these individuals still will likely have an elevated level of the CCR5 receptor protein at least initially, which may increase their risk of a hyperactive cytokine response. The down-regulation is possibly an attempt to reduce this risk, but it may not act quickly enough. This resting state difference in CCR5 protein levels should be explored as a potential predictor of COVID-19 infection severity.

We caution that while our experimental design was optimized for detecting *cis*-eQTLs variants effects and within a multi-potent immune-related cancer cell line, other longer range interactions between genomic regions and in other cell types may also be mediating severe COVID-19 response. For example, when comparing the response genes of the two identified *trans*-eQTLs in the introgressed haplotype to RNA-seq studies testing COVID-19 infection in lung cell lines and tissues (*Supplementary file 1h* and *Supplementary file 1i*), 5 (45%) and 14 (41.2%) of the response genes for the two *trans*-eQTLs in this locus rs13063635 and rs13098911, respectively, were detected as differentially expressed in at least one in vitro experiment, although neither of these variants were detected as emVars in this MPRA. We also stress that our *SARS-CoV-2* viral transfection reporter experiments represent only one clinically relevant context and that other emVars whose activity does increase upon infection may also be relevant. Therefore, we also urge additional functional studies to consider the effects of these *trans*-eQTLs and, in general, to replicate our findings on *cis*-eQTLS in other lung epithelial and other cell types from which some of the expression evidence we here build on are derived.

While our study provides strong experimental support for at least four archaic variants, and notably strong functional support for three variants (rs34041956, rs71327024, rs35454877) in the presence of viral transfection, at the introgressed locus, a deeper understanding of the regulatory architecture and the direction of the effect of these alleles in both the healthy and infected state and across different cell types needs further clarification. For example, in the severe COVID-19 phenotype, *CCR1* and *CCR5* are upregulated relative to their expression levels in moderate cases of the disease (*Chua et al., 2020*) and we saw evidence of upreglation by introgressed alleles in our mock infection experiment as well. However, in our MPRA experiments in K562 cells the three top candidate emVars acting as regulators of *CCR1* all showed down-regulation. Furthermore, alleles in Block A of the introgressed

haplotype, which exhibit the strongest GWAS associations with COVID-19 severity, all act as down-regulating eQTLs for *CCR1* in healthy whole blood samples. We hypothesize that this difference between the direction of effect of the alleles in healthy whole blood in vivo and K562 reporter assay in vitro episomal condition relative to in the A549-ACE2 disease state, reflects both differential regulation of these genes in different cell types and upon infection, and possibly that these alleles contribute to the risk of severe COVID-19 by destabilizing the regulatory mechanism of *CCR1* and *CCR5,* such that they have decreased expression in some cell types and conditions, but are hyper-expressed in others. Additional work needs to be done to further explore this potential mechanism as well as to uncover the molecular factors instrumental to it. Indeed, although hypothesizing a role for transcription factors IKZF1 or CEBPB (possibly interacting with another SNP at the locus), our TF binding site analysis could not identify a potential binding-altering mechanism for these variants.

In light of the efforts of multiple groups devoted to narrowing down the genes mediating the COVID-19 severity association, we conclude that several genes at this locus on chromosome 3, including *CCR1* and *CCR5*, display promising evidence for having a role in the underlying biological mechanisms. The involvement of most of them in the chemokine signaling pathway, and the evidence of coregulation provided by eQTL, epigenetic, and expression analyses, bring support to the hypothesis that the COVID-19 response is modulated in a concerted way.

## Methods

### Introgression scans

#### Sprime

We downloaded the Sprime software from https://faculty.washington.edu/browning/sprime.jar and ran it using java-v1.8.0_40. We used 423 Estonians as the ingroup population along with 36 African samples from the Simons Genome Diversity Project (SGDP) with no evidence of European admixture (*Mallick et al., 2016*) as an outgroup (*Supplementary file 1a*). Following *Browning et al., 2018*, we then summed the number of Sprime alleles per segment, and for segments greater than or equal to 30 alleles, we calculated the match rate to the Vindija Neanderthal genome (*Prüfer et al., 2017*) and the Denisovan genome (*Meyer et al., 2012*). Segments with greater than 0.6 match rate to the Neanderthal genome and less than 0.4 match rate to the Denisovan genome, were considered introgressed by Neanderthals. In total, we identified 693 such segments (*Supplementary file 1c*), including the segment containing the COVID-19 severity haplotype on chromosome 3 (see above).

#### U and Q95

Concerning U and Q95, following *Racimo et al., 2017*, for every 40 kb window within the genome, we calculated the U score as the number of SNPs per 40 kb window which had <1% frequency in a combined panel of African (AFR) individuals from the 1000 Genomes project (*Auton et al., 2015*) that show no major evidence of European admixture (*Supplementary file 1a*), had >20% frequency in 423 Estonians, and are homozygous in the Vindija Neanderthal genome (*Prüfer et al., 2017*). We also calculated the Q95 score as the 95% quantile frequency of the derived alleles in the Estonian set that are homozygous for the Vindija Neanderthal allele and are at <1% in the African set. We finally determined the top scoring windows to be those that are in the top 99th percentile of windows in terms of both U and Q95 scores (FDR 0–5.5%; *Racimo et al., 2017*).

#### GWAS data

GWAS data was downloaded from release 5 from the COVID-19 Host Genetics Institute (*COVID-19 Host Genetics Initiative, 2021*). We utilized the 'A2_ALL_leave_23andme' dataset for which the tested phenotype is 'Very severe respiratory confirmed covid vs. population'. There are 5,582 cases and 709,010 controls in this dataset.

#### eQTL data

eQTL data was obtained from the public repository of the eQTLGen consortium http://www.eqtlgen.org/cis-eqtls.html. Only significant *cis-* and *trans-*eQTLs, multiple hypothesis corrected p-value <0.05,

were included. Whenever Z-scores are reported including in *Supplementary file 1e-g*, the scores are polarized to the correct direction of effect of the allele along the introgressed haplotype.

## Covid-19 and related RNA-seq datasets

Datasets for RNA-seq studies performed on in vitro lung cell lines exposed to either *SARS-CoV-2* infection, related coronaviruses (e.g. MERS), other virus infection (e.g. RSV), or immune stimulation were obtained from the GEO database. Namely, GSE147507 provided by the tenOever lab (*Blanco-Melo et al., 2020*; Daemen et al. 20201) – Series 1–9 and 15, GSE139516 (*Zhang et al., 2020*), GSE122876 (*Yuan et al., 2019*), and GSE151513 (*Banerjee et al., 2021*). Raw RNA-seq data were all processed with a similar pipeline. Sequence read quality was checked with FastQC (https://www.bioinformatics.babraham.ac.uk/projects/fastqc/), with reads subsequently aligned to the human reference transcriptome (GRCh37.67) obtained from the ENSEMBL database (*Hunt et al., 2018*) which was indexed using the 'index' function of Salmon (version 0.14.0) (*Patro et al., 2017*) with a k-mer size of 31. Salmon alignment was performed with the 'quant' function with the following parameters: '-l A --numBootstraps 100 --gcBias --validateMappings'. All other parameters were left to defaults. Resulting quantification files were loaded into R (version 3.6.1) (*R Development Core Team, 2017*) via the tximport library (version 1.14.0; *Soneson et al., 2015*) with the 'type' option set to 'salmon'. Transcript quantifications were summarized to the gene level using the corresponding hg19 transcriptome GTF file mappings obtained from ENSEMBL. Count data were subsequently loaded into DESeq2 (version 1.26.0; *Love et al., 2014*) using the 'DESeqDataSetFromTximport' function. For subsequent differential-expression analysis, a low-count filter was applied prior to library normalization, wherein a gene must have had a count greater than five in at least three samples (in a given dataset) in order to be retained. For tenOever datasets, differential expression analysis was performed comparing treated samples (i.e., infected of stimulated cells) relative to the respective series' mock control samples. For the *Yuan et al., 2019* dataset, expression was compared between MERS-infected and mock controls. For the *Zhang et al., 2020* dataset, hours post-infection were used as a continuous variable (with mock representing '0 H post-infection') for the DESeq2 model, with significance defined as a gene being up- or down-regulated as a function of post-infection time. The differential expression analysis for the processed *Banerjee et al., 2021* dataset, which is also a time-course dataset, was implemented as described previously (*Banerjee et al., 2021*). Sets of significant genes in each dataset (defined as having a Benjamini-Hochberg adjusted p-value of <0.05) were subsequently intersected with the sets of response genes identified for the *cis*- and *trans*-eQTLs described in this study.

## MPRA design and implementation

We used an MPRA to determine which of these 613 variants within the introgressed segment fall within active CREs and whether they modulate reporter gene expression relative to the other variant at the same position. We conducted this assay in K562 cells, a leukemia cell line that displays multipotent hematopoietic biology and which allows for comparison between MPRA data and eQTL datasets derived from whole blood samples. Furthermore, K562 cells can be induced into cell fates associated with the COVID-19 phenotypes including monocyte and macrophage and neutrophils (*Butler and Hirano, 2014*).

Each variant was tested in the context of 270 bp of the endogenous sequence centered around each variant. For the Sprime alleles, if there is another allele within the span of this 270 bp sequence that is highly linked (r2 >0.8) in the Estonian population and at least in 9 of the 10 1KG populations it was included in the 270 bp sequence. This 270 bp sequence will be hereafter referred to as the 'tested sequence'. Additionally, we included 44 control sequences from a past MPRA experiment performed in K562 cells (*Jagoda et al., 2021*) with the 22 strongest up-regulating sequences from this MPRA serving as positive controls and the 22 sequences with smallest magnitude of effect on expression serving as negative controls. In the original assay, these control sequences were 170 bp in length, here we extended the sequences to create 270 bp sequences centered around the original 170 bp sequence. This difference in length could account for any potential regulatory discrepancy between the original experiment and this one. In total, the MPRA experiment consisted of 1,270 sequences. All tested sequences additionally included 15 bp of adaptor sequence on both the 5' and 3' side to

facilitate cloning into the MPRA vector. Following *Tewhey et al., 2016*, sequences were cloned into the MPRA vectors oriented according to the nearest transcription start site.

The tested sequences were synthesized by Twist Bioscience and the cloning steps to generate the MPRA vector library were conducted following the procedure outlined by *Tewhey et al., 2016* using the scale of their smaller library size. Barcoded sequences were initially cloned into pGL4:23:ΔxbaΔluc vectors and four sequencing libraries were prepared to sequence across the oligos and barcodes to determine oligo-barcode combinations within this mpraΔorf pool. Sequencing was conducted by the Harvard Bauer Core facility on an Illumina NovaSeq using 2x250 bp chemistry. Again, following *Tewhey et al., 2016*, an amplicon containing a GFP open reading frame, minimal promoter, and partial 3' UTR was cloned into the mpraΔorf library to make the final mpra:gfp library.

For each of four biological replicates, 40 μg of mpra:gfp vector pool was then transfected into 10 million K562 cells using electroporation with the Lonza 4D-Nucleofector following the manufacturer's protocol. After 24 hr, cells were collected and flash frozen in liquid $N_2$. Closely following *Tewhey et al., 2016* procedure for their smaller library, total RNA was extracted, GFP mRNA was isolated, converted to cDNA, and prepared into sequencing libraries to sequence the barcodes. Four sequencing libraries were also prepared of the mpra:gfp plasmid to obtain the representation of each barcode in the transfected vector pool. Barcode sequencing was performed on an Illumina MiSeq with 1x50 bp chemistry at the Harvard Bauer Core.

## MPRA data analysis

### Barcode - Oligo reconstruction

All MPRA data analysis steps were conducted following *Tewhey et al., 2016*. The 250 bp paired end reads from the sequencing of the mpraΔorf library were merged using Flash v.1.2.11 (*Magoč and Salzberg, 2011*). Merged amplicon sequences were then filtered for quality control such that sequences were kept if (1) there was a perfect match of 10 bp on the left or right side of the barcode, (2) the 10 bp on both sides of the barcode matched with levenshtein distance of 3 or less, and (3) the 2 bp on either side of the barcode matched perfectly. Sequences that passed through these filters were aligned back to the expected sequence pool using Bowtie2 v. 2.3.4.1 with the `--very-sensitive` flag. Alignments that had less than 95% perfect matching with the expected sequence and any alignment which had a mismatch at the variant position were removed. Barcodes that matched to more than one expected sequence are unusable and therefore were also removed.

Because of our small library size, we were able to get a very high barcode yield. All oligo sequences were represented and tagged with a wide diversity of barcodes, with the median unique barcodes per oligo being 18,812. Only five oligos had fewer than 100 unique barcodes, with the fewest being tagged by 27 unique barcodes. This large number of unique barcodes lead to extremely high reducibility between replicates (*Figure 2A*), which will allow for a high degree of sensitivity to detect subtle differences between alleles (*Tewhey et al., 2016*).

### Tag sequencing

Again, following *Tewhey et al., 2016*, the 1x50 bp tag sequencing reads were filtered such that reads were only kept if they had a maximum levenshtein distance of 4 with the constant sequence within the 3' UTR of the GFP as well as a perfect match with the two base pairs adjacent to the barcode. If the sequence passed through these filters, the barcodes were then matched back to the oligos based on the information from the mpraΔorf library sequencing described above. The counts for each barcode were summed for each oligo. This summation of the counts per barcode reduces the noise that could be derived from any individual barcode having a functional effect.

### Determination of active putative *cis*-regulatory elements and expression modulating variants (emVars)

Following *Tewhey et al., 2016*, the summed oligo counts from the tag sequencing for all four cDNA samples and all four plasmid samples were passed in DESeq2 and sequencing depth was normalized using the median-of-rations method (*Love et al., 2014*). We then used DESeq2 to model the normalized read counts for each oligo as a negative binomial distribution (NB). DESeq2 then estimates the variance for each NB by pooling all oligo counts across all the samples and modeling the relationship between oligo counts and the observed dispersion across all the data. It then estimates the

dispersion for each individual oligo by taking this observed relationship across all the data as a prior to performing a maximum posteriori estimate of the dispersion for each oligo. Therefore, the bias for the dispersion estimate for each oligo is greatly reduced because it relies on pooled information from all other oligos. We then used DESeq2 to estimate whether an oligo sequence had an effect on transcription by calculating the log fold change (LFC) between the oligo count in the cDNA replicates compared with its count in the plasmid pool. We tested whether this LFC constituted a significant difference of expression using Wald's test and required a stringent Bonferroni corrected p-value of less than 0.01 for a significant result. If an oligo sequence had a significant LFC with either allele, the sequence is considered '"active'. Finally, to determine which variants are expression modulating, for oligos which were determined to be active, we used DESeq2 to calculate the fold change between the two versions of the oligo sequences with Wald's test to calculate the p-values. p-values were then corrected using the Benjamini-Hochberg test to correct for multiple hypothesis testing. Significance was defined stringently as a multiple hypothesis corrected p-value of <0.01.

## emVar prioritization and intersection of data from other functional sources

To identify which of the 20 variants identified by the MPRA as emVars are the best candidates for contributing to an increased risk of severe COVID-19, we analyzed the MPRA data with data from other sources specifically:

## GWAS data

As described above, GWAS data is from a recent release (release 5) from the **COVID-19 Host Genetics Initiative, 2021**. If the p-value for an emVar was less than $5 * 10^{-8}$ (or -log10(p-value)>7.3), it was considered significant and passed through this prioritization step.

## Promoter-capture Hi-C datasets

Promoter-capture (Hi-C) data were obtained from **Jung et al., 2019**, particularly the file 'GSE86189_all_interaction.po.txt.gz' containing processed information on genomic regions with significant contacts of targeted promoters. This dataset was generated from promoter-capture assays across a number of different tissues and cell-types; given our particular interest in immune cell regulation, we considered only those significant interactions (reported p-value <0.01) present in samples from lymphoblasts (GM12878.ADS and GM19240.ADS), spleen (STL001.SX1 and STL003.SX3), and thymus (STL001.TH1) samples. Interacting regions, which may indicate putative CREs, were intersected with our defined emVars using bedtools intersect. If an emVar falls within a contact site for any gene, this is reported in **Table 1**. For our prioritization, if an emVar is both within a contact site for a gene with relevance to COVID-19 infection, particularly *CCR1*, which is differentially expressed in some *SARS-COV-2* infection studies (**Supplementary file 1h** and **Supplementary file 1i**) and by others (**Chua et al., 2020**), and *CCR5* which is reported as differentially expressed in other studies (**Chua et al., 2020**), and the emVar is also an eQTL for this same gene (**Võsa et al., 2018**), that supports its prioritization (**Table 1**).

## Candidate *cis*-regulatory elements (CCREs) by ENCODE

To further validate emVars for biological relevance, we downloaded all 926,535 human cCREs from https://screen.encodeproject.org/ (**Moore et al., 2020**). cCREs are DNAse hypersensitivity sites that are further supported by additional evidence of *cis*-regulatory activity in the form of either histone modifications (H3K4me3 and H3K27ac) or CTCF-binding data. These cCREs were further classified based on the combination of both their epigenomic signals as well as their genomic context into four major categories: cCREs with promoter-like signatures (cCRE-PLS), cCREs with enhancer-like signatures (cCRE-ELS, these are further subsetted as either proximal [pELS] or distal [dELS]), DNase-H3K4me3 cCREs, and CTCF-only.

To intersect these human cCREs with our emVar data, we first used LiftOver **Hinrichs et al., 2006** to convert our emVar coordinates from *GRCh37 to GRCh38* and then used Bedtools intersect to search for emVars falling within cCREs. These intersections are reported in **Table 1**. For emVars that already passed through the prior prioritization steps that overlapped a cCRE, we then examined the tissue level information on the cCRE on the web browser (https://screen.encodeproject.org/). emVars were

prioritized if they are within cCREs that had cCRE annotations in at least one immune-related 'class a' cell line, which is a cell line for which data on all four makers (DNAse, H3K4me3, H3K27ac, CTCF-binding) is available. These results are displayed in *Figure 2D* and *Table 1*.

## TF binding analysis

We used ENCODE (*Davis et al., 2018*) TF ChIPseq NarrowPeaks data (downloaded on 13/09/2021) for COVID19-relevant untreated conditions: blood-derived cell lines (K562,GM12878), lung-derived cell lines (A549, IMR-90), as well as primary cells from lung fibroblasts and upper lobe of left lung tissue experiments.

We extracted TF Positional Weight Matrices (PWMs) from HOCOMOCO v11 (*Kulakovskiy et al., 2018*) core human dataset and scored all positions within ChIPseq peaks with the PWM for the relevant TF following methods in *Molineris et al., 2013*. In particular, we defined TF affinity as log2 of the ratio between the probability of obtaining a motif from the TF PWM and the probability of obtaining it from a background model given by GC-content in the human genome. Considering only positions that overlap a SNP, and both the modern human and archaic alleles, we define putative TF binding sites (TFBS) as those with an affinity score above a cutoff defined as $-0.021*S_M^2 +1.020*S_M +0.032$, where $S_M$ is the maximal score for that PWM (after the local maximum at $S_M = 24.2857$ the cutoff is set to 12.4177). We define affinity-changing SNPs if only one of the alleles yields an affinity greater than such cutoff and their difference is greater than 0.5.

## Generation of custom plasmids for *SARS-CoV-2* virus infections

Four paired constructs (introgressed vs non-introgressed) were designed to check the activity of our four top emVars (rs71327024, rs71327057, rs34041956 of *CCR1* and rs35454877 of *CCR5*). Each oligonucleotide and the promoter fragment of the corresponding gene were subcloned into a vector (Promega Monster Green Fluorescent Protein phMGFP vector; catalog number E6421) containing EGFP. The custom plasmid synthesis was done via Genewiz, where the EGFP and the minimal promoter of the vector was replaced by each oligonucleotide +promoter of the corresponding gene (sequences of the eight custom plasmids are given in *Supplementary file 1m*). The eight custom plasmids referred to as A-I in the subsequent *SARS-CoV-2* virus infection experiments are listed as below: promoter of the target gene is given first, followed by the four SNP rsID and if the oligonucleotide is introgressed or not. A: CCR1_ rs71327024_non-introgressed; B: CCR1_ rs71327024_introgressed; C: CCR1_ rs71327057_non-introgressed; D: CCR1_ rs71327057_introgressed; E: CCR1_ rs34041956_ non-introgressed; F:CCR1_ rs34041956_introgressed; G: CCR5_ rs35454877_non-introgressed; H: CCR5_ rs35454877_introgressed; I: control plasmid (phMGFP vector).

## Cell culture and *SARS-CoV-2* virus infection

A549-ACE2 cells (human lung adenocarcinoma derived cells overexpressing ACE2) were generated by transducing A549 cells (ATCC) with lentiviruses carrying the ACE2 coding sequence. Lentiviruses were produced by co-transfection of HEK293T/17 cells with a transfer plasmid encoding ACE2 (Addgene #154981), the lentiviral packaging plasmid psPAX2 (Addgene #12260) and an envelope plasmid encoding VSV-G (Addgene #8454). Following transduction and selection with 10 μg/mL blasticidin, bulk cell populations were diluted to single cells by limiting dilution, and single-cell clonal populations were expanded. For this study, we used the clonal population A549-ACE2 B9. A549-ACE2 cells were maintained in Ham's F-12K (Kaighn's) medium supplemented with 10% fetal bovine serum (FBS; Sigma-Aldrich), 10 μg/ml blasticidin (Gibco), and Penicillin/Streptomycin (Pen/Strep; VWR). For *SARS-CoV-2* infections, $1.5x10^5$ cells/well were seeded in a 12-well plate for 24 hr. After 24 hr, cells were transfected with 200 ng of plasmids A-I (see above) for 24 hr, followed by mock infection or infection with ancestral *SARS-CoV-2* (VIDO-01 isolate) for 24 hr. After 24 hr of infection, bulk RNA was extracted from mock infected and *SARS-CoV-2*-infected cells, followed by gene expression analyses using quantitative real-time PCR (qRTPCR). *SARS-CoV-2* infections were performed at a multiplicity of infection (MOI) of 1. Regarding transfection controls, equal amounts of plasmid I (control plasmid, 200 ng) were transfected for both mock and *SARS-CoV-2*-infected cells. We noticed comparable amount of eGFP transcripts across all experimental conditions for plasmid I. This demonstrates that transfection efficiency did not vary significantly between experimental conditions in A549-ACE2 cells. Experiments,

performed in triplicate with *SARS-CoV-2,* were performed in a BSL3 laboratory at the Vaccine and Infectious Disease Organization, University of Saskatchewan following approved protocols.

## Quantitative real-time PCR (qRTPCR)

Bulk cellular RNA extraction was performed using RNeasy Mini Kit (Qiagen) according to the manufacturer's protocol. Four hundred nanograms of purified RNA was reverse transcribed using iScript gDNA Clear cDNA Synthesis Kit (Bio-Rad). Quantitative PCR reactions were performed with SsoFast EvaGreen supermix (Bio-Rad) for eGFP and *SARS-CoV-2* UpE Relative mRNA expression of eGFP was normalized to UpE and presented as 40-Ct. Primer sequences used were SARS2 UpE F – ATTGTTGA TGAGCCTGAAG, SARS2 UpE R – TTCGTACTCATCAGCTTG, eGFP F - ATGAAGGGTGTGGACG ACTG and eGFP R- CGCACGTACATCTTCTCGGT. qRTPCR to determine UpE levels was performed using as previously described (*Banerjee et al., 2017*).

# Acknowledgements

We thank Drs. Steve Reilly, Maryellen Ruvolo, David Pilbeam, Dan Lieberman, and David Reich, as well as members of the Capellini and Sabeti labs for their critical insight into this work. DM and LP are supported by the European Regional Development Fund, projects No. 2014–2020.4.01.16-0024, MOBTT53. FM is supported by the European Regional Development Fund, project No. 2014–2020.4.01.16-0030. Work in AB's laboratory is supported by research grants from Natural Sciences and Engineering Research Council of Canada (NSERC), Canadian Institutes of Health Research (CIHR), Saskatchewan Health Research Foundation (SHRF), and Coronavirus Variants Rapid Response Network (CoVaRR-Net). VIDO receives operational funding for its CL3 facility (InterVac) from the Canada Foundation for Innovation (CFI) through the Major Science Initiatives. VIDO also receives operational funding from the Government of Saskatchewan through Innovation Saskatchewan and the Ministry of Agriculture. EJ is supported NSF DDRIG (BCS-1847287) and TDC by NSF (BCS-2020205) and The American School of Prehistoric Research, Harvard University. VG is suproted by a postgraduate doctoral scholarship (PGS-D) funded by NSERC.

# Additional information

### Funding

| Funder | Grant reference number | Author |
| --- | --- | --- |
| European Regional Development Fund | 2014-2020.4.01.16-0024 | Luca Pagani<br>Davide Marnetto |
| European Regional Development Fund | 2014-2020.4.01.16-0030 | Francesco Montinaro |
| National Science Foundation | BCS-1847287 | Evelyn Jagoda |
| National Science Foundation | BCS-2020205 | Terence D Capellini |
| Natural Sciences and Engineering Research Council of Canada | | Arinjay Banerjee |
| Canadian Institutes of Health Research | | Arinjay Banerjee |
| Saskatchewan Health Research Foundation | | Arinjay Banerjee |
| Coronavirus Variants Rapid Response Network | | Arinjay Banerjee |
| Canada Foundation for Innovation | | Arinjay Banerjee |

| Funder | Grant reference number | Author |
| --- | --- | --- |
| Innovation Saskatchewan | | Arinjay Banerjee |
| Saskatchewan Ministry of Agriculture | | Arinjay Banerjee |

The funders had no role in study design, data collection and interpretation, or the decision to submit the work for publication.

## Author contributions

Evelyn Jagoda, Conceptualization, Data curation, Formal analysis, Validation, Investigation, Visualization, Methodology, Writing – original draft, Writing – review and editing; Davide Marnetto, Data curation, Formal analysis, Validation, Investigation, Visualization, Methodology, Writing – original draft, Writing – review and editing; Gayani Senevirathne, Formal analysis, Validation, Investigation, Visualization, Methodology, Writing – review and editing; Victoria Gonzalez, Formal analysis, Investigation, Visualization, Methodology; Kaushal Baid, Investigation, Methodology; Francesco Montinaro, Formal analysis, Validation, Investigation, Writing – review and editing; Daniel Richard, Data curation, Formal analysis, Writing – review and editing; Darryl Falzarano, Emmanuelle V LeBlanc, Che C Colpitts, Investigation; Arinjay Banerjee, Supervision, Validation, Investigation, Methodology, Writing – review and editing; Luca Pagani, Formal analysis, Supervision, Investigation, Writing – original draft, Writing – review and editing; Terence D Capellini, Conceptualization, Data curation, Supervision, Funding acquisition, Investigation, Writing – original draft, Project administration, Writing – review and editing

## Author ORCIDs

Che C Colpitts http://orcid.org/0000-0003-2474-1834
Arinjay Banerjee http://orcid.org/0000-0002-2821-8357
Terence D Capellini http://orcid.org/0000-0003-3842-8478

## Decision letter and Author response

Decision letter https://doi.org/10.7554/eLife.71235.sa1
Author response https://doi.org/10.7554/eLife.71235.sa2

---

# Additional files

## Supplementary files

• Supplementary file 1. This is an excel file containing supplementary data in the form of 13 tables/sheets called (a-m). (a) presents the African samples used as outgroup for our Sprime introgression scan. (b) presents the full results from our Sprime scan. (c) presents Neanderthal matching Sprime segments. (d) presents results from our U and Q95 scan. (e) presents cis-eQTLs from our U and Q95 Scan. (f) presents cis-eQTLs from our Sprime scan. (g) presents trans-eQTLs from our Sprime scan. (h) presents expression of genes responsive to cis-eQTLs within the introgressed region in RNA-seq studies of infection by SARS-CoV-2 and related viruses. (i) presents expression of genes responsive to trans-eQTLs within the introgressed region in RNA-seq studies of infection by SARS-CoV-2 and related viruses. (j) presents data on all MPRA tested variants. (l) presents data from our ChIP-seq analyses. (l) presents our analyses on ChIP-seq data and predicted transcription factor binding sites (TFBS). (m) presents our construct sequences used in transfection studies.

• Transparent reporting form

## Data availability

The data supporting the findings of this study, as presented in all figures (Figures 1-5), including Figure Supplements, are available within Supplementary File 1 containing 13 sheets (a-m) included in this publication as well as from the corresponding author upon reasonable request. We have deposited all MPRA data in Geo Omnibus (GSE176233).

The following dataset was generated:

| Author(s) | Year | Dataset title | Dataset URL | Database and Identifier |
|---|---|---|---|---|
| Jagoda E, Marnetto D, Senevirathne G, Gonzalez V, Baid K, Montinaro F, Richard D, Falzarano D, LeBlanc EV, Colpitts CC, Banerjee A, Pagani L, Capellini TD | 2023 | Regulatory dissection of the severe COVID-19 risk locus introgressed by Neanderthals | https://www.ncbi.nlm.nih.gov/geo/query/acc.cgi?acc=GSE176233 | NCBI Gene Expression Omnibus, GSE176233 |

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
