## [Editor Report]

A genetic haplotype on chromosome 3 that entered the human lineage from mating with Neanderthals has previously been implicated as a strong genetic risk factor for severe COVID-19 outcomes. This study uses population genetics and functional genomics tools along with experimental assays to assess the genetic variants in these regions for their likelihood of driving the severe COVID-19 phenotype. They ultimately identify 4 (out of about 600) variants as strong functional candidates. This study is a valuable contribution to the interaction between host genomics and COVID-19 outcomes and provides compelling evidence allowing for more targeted future functional investigations.

---

## [Decision Letter]

**Decision letter after peer review:**

Thank you for submitting your article "Regulatory dissection of the severe COVID-19 risk locus introgressed by Neanderthals" for consideration by *eLife*. Your article has been reviewed by 3 peer reviewers, and the evaluation has been overseen by a Reviewing Editor and Molly Przeworski as the Senior Editor. The reviewers have opted to remain anonymous.

Essential revisions:

All reviewers agree that the topic of the paper, identifying genetic features that contribute to covid-19 severity in a neanderthal introgressed region, is of general interest and the authors impressively combine various annotation and functional assessments of the variants. However, there are a range of concerns about the experimental and functional genomics evidence presented, leading to the conclusion that whether chemokine receptor expression is linked to the clinical phenotype remains unaddressed by the study. The majors concerns are:

(1) The first concern is the assumption that regulation of GFP expression under a minimal promoter correlates perfectly with regulation of CCR1 and CCR5 under their native promoter. The authors could simply address this in the text, but this will not address the underlying issue. Preferably, having narrowed their regions of interest down to 4 emVars, they would place these on a reporter construct with GFP under control of the CCR1 or CCR5 promoters (or a minimal section of these specific promoters) and repeat either their quantitative PCR analysis or, in this case, a fluorescence-based assay may suffice. This would greatly strengthen the argument that CCR1 and CCR5 are specifically regulated by these emVars. If they believe that an infectious context might change these results, they could conduct this reporter assay during viral infection using viruses that do not require BSL3 containment, or collaborate with a lab able to use SARS-CoV-2.

(2) The second issue is, as acknowledged by the authors, that the emVars they identify using computational approaches are downregulatory whereas CCR1 and CCR5 are upregulated during severe COVID-19. The authors included some speculation about how these regulatory regions might still be linked to upregulation of the chemokine receptors in a clinical context, but this closing section of the manuscript isn't very strong. The assay described above would be informative as to whether these emVars do in fact downregulate CCR1 and CCR5, and whether they do so during infection. As a further suggestion to address the discrepancy between clinical gene expression data and their MPRA data, the authors could perform the conventional reporter assay (such as luciferase assay) in epithelial or macrophage cell lines (not K562). At least they might want to test the four variants they highlighted in the manuscript.

(3) Enhancer activity should be defined in a relative manner, by comparing to the background activity of negative control sequences.

(4) The authors should propose the molecular mechanisms of the SNPs' function that underlie COVID19 severity. For example, the authors could perform TF motif search around the potential causative SNPs they found, see if the SNPs alter the TF motifs, and see if the alteration of the TF motifs can explain the eQTL effect direction.

*Reviewer #1 (Recommendations for the authors):*

There are, in my opinion, two primary weaknesses of the paper. The first is the assumption that regulation of GFP expression under a minimal promoter correlates perfectly with regulation of CCR1 and CCR5 under their native promoter. The authors could simply address this in the text, but this will not address the underlying issue. Preferably, having narrowed their regions of interest down to 4 emVars, they would place these on a reporter construct with GFP under control of the CCR1 or CCR5 promoters (or a minimal section of these specific promoters) and repeat either their quantitative PCR analysis or, in this case, a fluorescence-based assay may suffice. This would greatly strengthen the argument that CCR1 and CCR5 are specifically regulated by these emVars.

If they believe that an infectious context might change these results, they could conduct this reporter assay during viral infection using viruses that do not require BSL3 containment, or collaborate with a lab able to use SARS-CoV-2.

The second issues is, as acknowledged by the authors, that the emVars they identify using computational approaches are downregulatory whereas CCR1 and CCR5 are upregulated during severe COVID-19. The authors included some speculation about how these regulatory regions might still be linked to upregulation of the chemokine receptors in a clinical context, but this closing section of the manuscript isn't very strong. The assay described above would be informative as to whether these emVars do in fact downregulate CCR1 and CCR5, and whether they do so during infection. Depending on the results the authors may not need the very speculative hypothesis at the end of the paper.

*Reviewer #2 (Recommendations for the authors):*

1. The authors defined the activity of enhancers based on the log fold change (LFC) between barcode counts in cDNA comparing to those in plasmid library (p<0.01). With this definition, they claimed that 53% of the tested sequences had enhancer activity (Figure 2D). However, I do not think such absolute values (i.e. LFC cDNA count/pDNA count, p>0.01) can be considered as "activity" of enhancers. Rather, enhancer activity should be defied in a relative manner, by comparing to the background activity of negative control sequences. For example, Inoue et al. (Cell Stem Cell, 2019) identified active enhancers whose activity significantly deviated from that of the negative control sequences. Moreover, in the most case of traditional luciferase assays, researchers specify the activity of enhancers relative to that of a negative control (e.g. empty vectors etc.). Hence, I would like the authors to provide and visualize the following information:

(1) How many sequences (or what fraction) show significantly higher barcode expression (LFC cDNA/pDNA) than negative controls? These should be considered as "active enhancers", and the distribution of these relative activities can be visualized with a violin (or box) plot, comparing to the activity of negative and positive controls.

(2) Do these "active enhancers" significantly overlap with enhancer epigenetic marks (chromHMM enhancers, DHS, and worth including H3K27ac). It is similar to Figure 3B, but I expect the result will be improved.

(3) How many emVars fall into the "active enhancers"? Do the four eQTLs that are linked with CCR1/5 overlap with "active enhancers"? emVars that are not overlap with "active enhancers" should be less important, because the differential expression between the two alleles should be within the range of background noise.

2. The authors systematically identified potential candidates of causative SNPs associated with COVID19 severity, while previous GWAS only found a tag SNP. These findings could be much strengthened and warranted if the authors propose the molecular mechanisms of the SNPs' function that underlie COVID19 severity. For example, the authors could perform TF motif search around the potential causative SNPs they found, see if the SNPs alter the TF motifs, and see if the alteration of the TF motifs can explain the eQTL effect direction. They could further test if overexpression/knockdown of the candidate TFs affect CCR1/5 expression.

*Reviewer #3 (Recommendations for the authors):*

1. Schematic representation of computational analysis overview (like Vosa et al. 2018 Figure 1) would be helpful to understand the analysis flow and MPRA experiment design.

Computational analysis parts in the manuscript are descriptive and hard to follow up. Also, some important SNPs (rs13063635 and rs13098911) are not indicated in figure 1.

If rs13063635 and rs13098911 are in the same region shown in Figure 1, the author should indicate the position.

2. Now author narrowed down the four critical variants. To filling the gap of the discrepancy between clinical gene expression data and their MPRA data, this reviewer suggests the conventional reporter assay (such as luciferase assay) in epithelial or macrophage cell lines (not K562).

At least they might want to test the four variants they highlighted in the manuscript.

3. Hypothesis described in the discussion page #2 line 15~ is obscure and not great for the conclusion. Please make it clear.

4. This reviewer is curious if there are any common transcription factor binding sites around the 20 critical variants. If the author has any data of TF motif analysis would be nice as an additional figure.

---

## [Author Response]

Essential revisions:All reviewers agree that the topic of the paper, identifying genetic features that contribute to covid-19 severity in a neanderthal introgressed region, is of general interest and the authors impressively combine various annotation and functional assessments of the variants. However, there are a range of concerns about the experimental and functional genomics evidence presented, leading to the conclusion that whether chemokine receptor expression is linked to the clinical phenotype remains unaddressed by the study. The majors concerns are:(1) The first concern is the assumption that regulation of GFP expression under a minimal promoter correlates perfectly with regulation of CCR1 and CCR5 under their native promoter. The authors could simply address this in the text, but this will not address the underlying issue. Preferably, having narrowed their regions of interest down to 4 emVars, they would place these on a reporter construct with GFP under control of the CCR1 or CCR5 promoters (or a minimal section of these specific promoters) and repeat either their quantitative PCR analysis or, in this case, a fluorescence-based assay may suffice. This would greatly strengthen the argument that CCR1 and CCR5 are specifically regulated by these emVars. If they believe that an infectious context might change these results, they could conduct this reporter assay during viral infection using viruses that do not require BSL3 containment, or collaborate with a lab able to use SARS-CoV-2.

We thank the Editor and Reviewers for suggesting this experiment. For each of the four emVars we have now cloned the introgressed or non-introgressed variant emVar oligo upstream of each specific targeted promoter (e.g., *CCR1* or *CCR5*), upstream of eGFP in the targeting vector. These have now been transfected into cells in both the presence as well as absence of *SARS-CoV-2* by our colleagues (Dr. Arinajay Banerjee et al.) who have BSL3 containment. In this context, we had tried a number of different cell lines including the original K562 cell line used in our MPRA and Single Locus Reporter Assay experiments. However, K562 cells were not capable of being infected by the virus even at extremely low titers. Other cell lines either had similar issues or were killed upon low titer viral infection. As a result, we switched to another cell line routinely used to study *SARS-CoV-2* and other SARs and MERs infections, and one relevant to COVID-19 disease phenotypes: human lung epithelial (A549) cells that were engineered to express the receptor of *SARS-CoV-2*, angiotensin-converting enzyme 2 (ACE2). Upon performing the transfections of each emVar variant construct in the presence and absence of *SARS-CoV-2* infection, we were able to identify that three emVars continued to act as response variants as assessed using quantitative PCR (please see new Methods section with our updated experimental strategy for this section as well as new Supplementary File 1m with resulting construct details). These new findings are presented as a separate new Results section entitled “Functional Reporter Analyses of top 4 emVars Reveals Causal Variant Activity in the Presence of *SARS-CoV-2*” in the newly revised manuscript:

Additionally, we also updated the Discussion section to reflect the insights acquired from these experiments.

“We next tested these four emVars in reporter assays in a lung cell line (A549-ACE2) capable of expressing ACE2, the receptor for *SARS-CoV-2*, and in the presence and absence of *SARS-CoV-2*. […] This resting state difference in CCR5 protein levels should be explored as a potential predictor of COVID-19 infection severity.”

Additional, related text is present in the Discussion and Methods sections.

(2) The second issue is, as acknowledged by the authors, that the emVars they identify using computational approaches are downregulatory whereas CCR1 and CCR5 are upregulated during severe COVID-19. The authors included some speculation about how these regulatory regions might still be linked to upregulation of the chemokine receptors in a clinical context, but this closing section of the manuscript isn't very strong. The assay described above would be informative as to whether these emVars do in fact downregulate CCR1 and CCR5, and whether they do so during infection. As a further suggestion to address the discrepancy between clinical gene expression data and their MPRA data, the authors could perform the conventional reporter assay (such as luciferase assay) in epithelial or macrophage cell lines (not K562). At least they might want to test the four variants they highlighted in the manuscript.

We ask the Editor/Reviewers to please see our comments listed above in point (#1) and associated text. We also further updated our Discussion section to address this point and the overall strengths and weaknesses of the experimental approach.

Discussion Text:

“We caution that while our experimental design was optimized for detecting *cis*-eQTLs variants effects and within a multi-potent immune-related cancer cell line, other, longer range interactions between genomic regions and in other cell types may also be mediating severe COVID-19 response. […] The involvement of most of them in the chemokine signaling pathway, and the evidence of coregulation provided by eQTL, epigenetic and expression analyses, bring support to the hypothesis that the COVID-19 response is modulated in a concerted way.”

(3) Enhancer activity should be defined in a relative manner, by comparing to the background activity of negative control sequences.

Please see our comments to point #1 above in our newly revised manuscript highlighting more detailed functional experiments.

(4) The authors should propose the molecular mechanisms of the SNPs' function that underlie COVID19 severity. For example, the authors could perform TF motif search around the potential causative SNPs they found, see if the SNPs alter the TF motifs, and see if the alteration of the TF motifs can explain the eQTL effect direction.

The suggested analysis was performed as per Reviewers #2 and #3 suggestions. Although it could not suggest a straightforward molecular mechanism for the shortlisted variants, all resulting data are now included in the Results and associated tables. Please see the replies to the reviewer’s comments.

Reviewer #1 (Recommendations for the authors):There are, in my opinion, two primary weaknesses of the paper. The first is the assumption that regulation of GFP expression under a minimal promoter correlates perfectly with regulation of CCR1 and CCR5 under their native promoter. The authors could simply address this in the text, but this will not address the underlying issue. Preferably, having narrowed their regions of interest down to 4 emVars, they would place these on a reporter construct with GFP under control of the CCR1 or CCR5 promoters (or a minimal section of these specific promoters) and repeat either their quantitative PCR analysis or, in this case, a fluorescence-based assay may suffice. This would greatly strengthen the argument that CCR1 and CCR5 are specifically regulated by these emVars.

We have now performed these experiments as addressed above in the section addressed to the Editor (point number 1).

If they believe that an infectious context might change these results, they could conduct this reporter assay during viral infection using viruses that do not require BSL3 containment, or collaborate with a lab able to use SARS-CoV-2.

We have now performed these experiments as addressed above in the section addressed to the Editor (point number 1).

The second issues is, as acknowledged by the authors, that the emVars they identify using computational approaches are downregulatory whereas CCR1 and CCR5 are upregulated during severe COVID-19. The authors included some speculation about how these regulatory regions might still be linked to upregulation of the chemokine receptors in a clinical context, but this closing section of the manuscript isn't very strong. The assay described above would be informative as to whether these emVars do in fact downregulate CCR1 and CCR5, and whether they do so during infection. Depending on the results the authors may not need the very speculative hypothesis at the end of the paper.

We feel we have readdressed this important point in the newly revised Discussion section which is present in the updated manuscript.

Overall, we thank Reviewer #1 for their terrific comments as we believe their suggested experiments have now strengthened our findings and the importance of the candidate emVars as functional variants resulting from Neandertal introgression.

Reviewer #2 (Recommendations for the authors):1. The authors defined the activity of enhancers based on the log fold change (LFC) between barcode counts in cDNA comparing to those in plasmid library (p<0.01). With this definition, they claimed that 53% of the tested sequences had enhancer activity (Figure 2D). However, I do not think such absolute values (i.e. LFC cDNA count/pDNA count, p>0.01) can be considered as "activity" of enhancers. Rather, enhancer activity should be defied in a relative manner, by comparing to the background activity of negative control sequences. For example, Inoue et al. (Cell Stem Cell, 2019) identified active enhancers whose activity significantly deviated from that of the negative control sequences. Moreover, in the most case of traditional luciferase assays, researchers specify the activity of enhancers relative to that of a negative control (e.g. empty vectors etc.). Hence, I would like the authors to provide and visualize the following information:(1) How many sequences (or what fraction) show significantly higher barcode expression (LFC cDNA/pDNA) than negative controls? These should be considered as "active enhancers", and the distribution of these relative activities can be visualized with a violin (or box) plot, comparing to the activity of negative and positive controls.

We thank the reviewer for their insightful comment related to the point above. Consistent with several other MPRA papers, including the one from which our design was informed (Tewhey et al. 2016), we defined active enhancers based on the relative relation of each element’s cDNA/pDNA count rather than normalizing to the negative controls. The negative controls in this experiment were taken from a prior MPRA experiment we performed, however, the activity in this assay may be somewhat different because these elements were 270bp whereas in the prior experiment they were only 170bp. The inclusion of additional 100bp of potential regulatory sequence may affect the overall regulatory activity, therefore making these elements less than ideal for normalization of the experimental set as we cannot fully confidently discount their activity. However, in Author response image 1 we have provided the violin plots as requested. The dashed horizontal lines in the plot show the minimum positive and negative expression changes ascribed to active enhancers as we defined them. As you can see almost all the negative controls have a much smaller effect, with the exception of 3 outlier sequences that were denoted as active in our analysis. We attribute the activity of these negative controls to the addition of the 100bp sequence that was not included in the original assay in which we tested these sequences. Furthermore, the rightmost violin shows the LFC for sequences with significant skew – our 20 emVar sequences which we flagged for further analysis. Again they all have higher LFC than vast majority of negative control sequences. Furthermore, as our paper now integrates follow up experimental analysis on the emVars that we considered the most likely candidates based on the MPRA and subsequent analysis, and we are therefore not only relying on these MPRA analyses, we have an additional avenue of experimental support for the variants that we highlight, reducing the impact of these particular calculations.

**Author response image 1. sa2fig1:** 

(2) Do these "active enhancers" significantly overlap with enhancer epigenetic marks (chromHMM enhancers, DHS, and worth including H3K27ac). It is similar to Figure 3B, but I expect the result will be improved.

While we have kept our original definition of active enhancers, we took the reviewer’s suggestion and have updated Figure 4B (originally 3B) to now include H3K27ac as well.

(3) How many emVars fall into the "active enhancers"? Do the four eQTLs that are linked with CCR1/5 overlap with "active enhancers"? emVars that are not overlap with "active enhancers" should be less important, because the differential expression between the two alleles should be within the range of background noise.

Expression differences between the two alleles were only considered for variants in enhancers that were already determined to be active based on our original analysis. However, as shown in the violin plot above (addressing point (1)) they all have much greater LFC than almost all the negative control sequences.

2. The authors systematically identified potential candidates of causative SNPs associated with COVID19 severity, while previous GWAS only found a tag SNP. These findings could be much strengthened and warranted if the authors propose the molecular mechanisms of the SNPs' function that underlie COVID19 severity. For example, the authors could perform TF motif search around the potential causative SNPs they found, see if the SNPs alter the TF motifs, and see if the alteration of the TF motifs can explain the eQTL effect direction. They could further test if overexpression/knockdown of the candidate TFs affect CCR1/5 expression.

We thank Reviewer #2 for suggesting this analysis. We now explored TF binding across the whole locus using both binding site prediction through PWM motif search and empirical evidence of binding provided by ENCODE ChIP-seq experiments database. Predicted binding sites though TF motifs are extremely frequent throughout the genome, and this locus is no exception: we predicted 1428 binding events overlapping 388 SNPs of the greater than six hundred variants tested. We therefore restricted our analysis to those backed by empirical evidence.

While we could find several binding peaks overlapping our shortlisted SNPs (as many as 89 and 36 for rs71327024 and rs35454877 respectively, see Supplementary File 1k for full results) we could not highlight any changes in binding affinity for TFs with an empirical support at these sites. Nevertheless, this analysis brought to our attention a TF affinity-altering SNP: rs17713054 (see Supplementary File 1l, Figure 2—figure supplement 2). Although not being classified as emVar in our experimental setting and therefore not further investigated in our study, this SNP should be considered for future inspection. See also the reply to comment 4 of Reviewer #3.

Reviewer #3 (Recommendations for the authors):1. Schematic representation of computational analysis overview (like Vosa et al. 2018 Figure 1) would be helpful to understand the analysis flow and MPRA experiment design.

We thank the reviewer for this suggestion and have created this schematic as our new Figure 1.

Computational analysis parts in the manuscript are descriptive and hard to follow up. Also, some important SNPs (rs13063635 and rs13098911) are not indicated in figure 1.If rs13063635 and rs13098911 are in the same region shown in Figure 1, the author should indicate the position.

We have fixed the manuscript at a number of locations and updated the callouts in Fig2 to address the most recent findings based on new papers as well as our analyses.

2. Now author narrowed down the four critical variants. To filling the gap of the discrepancy between clinical gene expression data and their MPRA data, this reviewer suggests the conventional reporter assay (such as luciferase assay) in epithelial or macrophage cell lines (not K562).At least they might want to test the four variants they highlighted in the manuscript.

We thanks this Reviewer for this suggestion. For reasons outlined extensively above in our address to the Editor, we have now performed related transfection experiments for the four variants and in the presence/absence of *SARS-CoV-2* viral infection. Please see the Editor’s section above.

3. Hypothesis described in the discussion page #2 line 15~ is obscure and not great for the conclusion. Please make it clear.

We have now updated the Discussion to reflect the new experiments. We hope that we have more directly addressed this and all Reviewer points more clearly.

4. This reviewer is curious if there are any common transcription factor binding sites around the 20 critical variants. If the author has any data of TF motif analysis would be nice as an additional figure.

We thank the Reviewer for this suggestion. Also prompted by Reviewer #2 comment, we analyzed TF binding across the whole locus, both through TF PWM motif prediction and empirical evidence of binding provided by ENCODE ChIP-seq experiments database. We could find several binding peaks overlapping our shortlisted SNPs, finding IKZF1 to bind 7 of the 20 critical variants and all four prioritized SNPs, see Supplementary File 1k for full results. While refraining from producing data and relative figure about predicted TF binding events that were not supported by empirical evidence (see considerations made about comment 2 of Reviewer #2) we did produce an additional figure (Figure 2—figure supplement 2) describing the most interesting TF affinity-altering SNP: rs17713054.

Overall, we thank Reviewer #3 for their terrific comments.